# KIND: Knowledge Integration and Diversion for Training Decomposable Models

Yucheng Xie [1 2]  Fu Feng [1 2]  Ruixiao Shi [1 2]  Jing Wang [1 2]  Yong Rui [3]  Xin Geng [1 2]

## Abstract

Pre-trained models have become the preferred backbone due to the increasing complexity of model parameters. However, traditional pre-trained models often face deployment challenges due to their fixed sizes, and are prone to negative transfer when discrepancies arise between training tasks and target tasks. To address this, we propose **KIND**, a novel pre-training method designed to construct decomposable models. KIND integrates knowledge by incorporating Singular Value Decomposition (SVD) as a structural constraint, with each basic component represented as a combination of a column vector, singular value, and row vector from $U$, $\Sigma$, and $V^\top$ matrices. These components are categorized into **learngenes** for encapsulating class-agnostic knowledge and **tailors** for capturing class-specific knowledge, with knowledge diversion facilitated by a class gate mechanism during training. Extensive experiments demonstrate that models pre-trained with KIND can be decomposed into learngenes and tailors, which can be adaptively recombined for diverse resource-constrained deployments. Moreover, for tasks with large domain shifts, transferring only learngenes with task-agnostic knowledge, when combined with randomly initialized tailors, effectively mitigates domain shifts. Code will be made available at https://github.com/Te4P0t/KIND.

## 1. Introduction

The increasing size of models has significantly increased computational costs, making pre-trained models a corner-

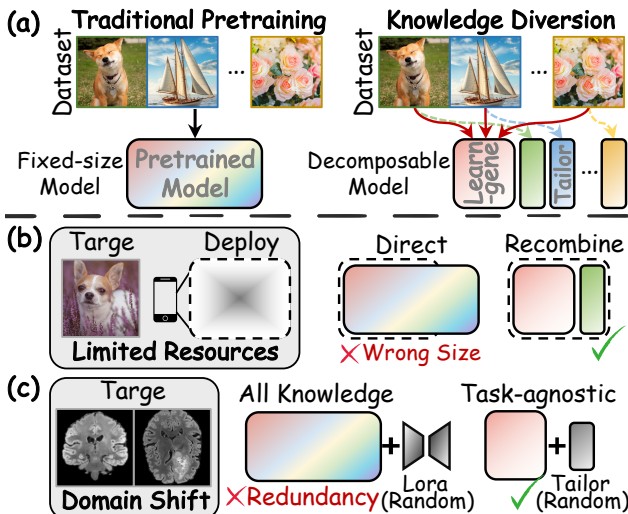

Figure 1. (a) Traditional pre-training prioritizes maximizing performance on training datasets, often producing fixed-size models and making them prone to negative transfer. In contrast, KIND redefines the training objective to pre-train models that are both structure- and knowledge-decomposable. (b) Consequently, KIND enables pre-trained models to be adaptively restructured, facilitating deployment in diverse resource-constrained scenarios. (c) Additionally, the task-agnostic knowledge encapsulated in learngenes can effectively mitigate domain shifts.

stone of modern machine learning (Qiu et al., 2020; Han et al., 2021; Feng et al., 2025b). These pre-trained models have proven highly effective, especially when combined with parameter-efficient fine-tuning (PEFT) techniques such as LoRA (Hu et al., 2022; Hayou et al., 2024) and its variants (Zhang et al., 2023; Valipour et al., 2023; Liu et al., 2024). However, traditional pre-training approaches primarily focus on optimizing performance for specific training datasets, often neglecting their transferability to downstream tasks and adaptability to diverse deployment scenarios.

As a result, pre-trained models typically have a fixed, large size, designed to encapsulate as much knowledge as possible from the training data. This design, however, presents significant challenges for practical deployment, which is often constrained by factors like memory usage, processing power, and response time (Zhang et al., 2022). More importantly, when downstream tasks differ significantly from the pre-training datasets, the transferred knowledge can

[1]School of Computer Science and Engineering, Southeast University, Nanjing, China [2]Key Laboratory of New Generation Artificial Intelligence Technology and Its Interdisciplinary Applications (Southeast University), Ministry of Education, China [3]Lenovo Research. Correspondence to: Jing Wang <wangjing91@seu.edu.cn>, Xin Geng <xgeng@seu.edu.cn>.

*Proceedings of the 42$^{nd}$ International Conference on Machine Learning*, Vancouver, Canada. PMLR 267, 2025. Copyright 2025 by the author(s).

become redundant (Feng et al., 2024), biased (Ren et al., 2024), or even harmful (Wang et al., 2019; Rosenstein et al., 2005). These limitations underscore that traditional pre-trained models may not always serve as optimal backbones, as illustrated in Figure 1. This raises a critical question: *Can we rethink the pre-training process to develop **decomposable pre-trained models** that can be adaptively adjusted to meet the specific requirements of downstream tasks and deployment scenarios?*

Recently, a novel knowledge transfer framework called *Learngene* has been introduced (Wang et al., 2023). Unlike traditional transfer learning methods, *Learngene* encapsulates task-agnostic knowledge into modular network fragments (Feng et al., 2023) known as learngenes, to enhance the efficiency of knowledge transfer and improve network adaptability. Building upon the *Learngene* framework, we propose KIND, a novel pre-training method that performs **K**nowledge **IN**tegration and **D**iversion during the pre-training process. KIND is designed to construct flexible and decomposable pre-trained models, facilitating adaptive transformations to address the diverse requirements of downstream tasks and deployment scenarios.

KIND decomposes the weight matrix into *basic components* for knowledge integration, then associates class-specific and class-agnostic knowledge with distinct components to facilitate knowledge diversion. For this decomposition, KIND employs Singular Value Decomposition (SVD), representing each basic component as a combination of a column vector, singular value, and row vector derived from the $U$, $\Sigma$, and $V^\top$ matrices. These basic components are categorized into two types: **learngenes**, which encapsulate class-agnostic knowledge, and **tailors**, which capture class-specific knowledge. Instead of directly applying SVD to pre-trained model weights (Han et al., 2023; Zhang & Pilanci, 2024; Robb et al., 2020), KIND incorporates SVD as a structural constraint during pre-training and trains the basic components rather than the full weight matrices. Such indirect training enables more explicit control over each class-specific component, guided by a class gate mechanism, thereby facilitating effective knowledge diversion.

We conduct experiments on class-conditional image generation tasks to better demonstrate knowledge transfer, using Diffusion Transformers (DiTs) (Peebles & Xie, 2023) as the backbone for diffusion models. We pre-train DiT-B and DiT-L with KIND on ImageNet-1K, resulting in decomposable models that can be effectively divided into learngenes and tailors. Extensive experiments evaluate KIND across three scenarios. 1) **General Tasks**: Models pre-trained with KIND perform on par with traditional pre-trained models (often outperforming them) without additional computational costs. 2) **Resource-constrained Scenarios**: KIND facilitates flexible combinations of learngenes and tailors

to meet storage and computational limits, maintaining performance without sacrificing performance. 3) **Tasks with Large Domain Shifts**: KIND transfers learngenes only, combined with randomly initialized tailors, enabling efficient adaptation via class-agnostic knowledge.

Our main contributions are as follows: 1) We redefine the pre-training objective by shifting the focus from solely maximizing model performance to diverting knowledge into class-agnostic knowledge and class-specific components, facilitating the construction of a more flexible and decomposable backbone adaptable to various scenarios. 2) We propose KIND, a novel pre-training method that integrates and diverts knowledge, marking the first application of learngenes to image generation tasks. 3) We establish a new benchmark for evaluating transfer efficiency and flexibility in diffusion models. Extensive experiments demonstrate that KIND achieves state-of-the-art performance while providing flexible storage and computational efficiency.

## 2. Related Work

### 2.1. Initialization and Training of Variable-sized Models

Practical deployments often encounter constraints related to memory usage, processing power, and response time, necessitating models of variable sizes (Zhang et al., 2022). However, traditional pre-trained models are typically fixed in size, requiring ***retraining*** when a suitable model size is unavailable (Qiu et al., 2020; Han et al., 2021). While traditional model compression techniques, such as knowledge distillation (Gou J, 2021; Muralidharan et al., 2024) and model pruning (Zhang et al., 2024a; Castells et al., 2024), can generate models of variable sizes, they involve ***repeated operations*** for each model size, resulting in significant inefficiencies in both time and resource consumption.

The *Learngene* framework, inspired by the transfer of genetic information in nature (Feng et al., 2023), encapsulates common knowledge into modular network fragments, termed "learngenes", and employs them to initialize variable-sized models (Wang et al., 2023). Notably, the process of condensing knowledge from pre-trained models into learngenes incurs a ***one-time cost***, eliminating the need for further training during model initialization. Current learngene-based methods, either direct transfer selected layers from pre-trained models (Wang et al., 2022; 2023), or apply predefined rules (e.g., Kronecker products) to distill knowledge knowledge into learngenes (Xia et al., 2024; Feng et al., 2025a). However, these approaches neglect the alignment between model components and their corresponding knowledge, limiting their efficiency and adaptability.

In contrast, KIND enhances such alignment through knowledge diversion during pre-training, constructing a decomposable model that enables more flexible and efficient ini-

tialization across varying model sizes.

## 2.2. Parameter Efficient Fine-Tuning (PEFT)

The increasing complexity of model parameters has made fine-tuning all parameters of pre-trained models resource-intensive and time-consuming (Touvron et al., 2021; Achiam et al., 2023). To address this, PEFT techniques are developed to adapt large pre-trained models to new tasks by fine-tuning only a small set of parameters (Hu et al., 2022; Houlsby et al., 2019; Hu et al., 2023; Chen et al., 2022). Recent approaches apply SVD to pre-trained weight matrices, fine-tuning models by adjusting singular values, a process known as spectral shift (Han et al., 2023; Robb et al., 2020; Sun et al., 2022), or by fine-tuning singular vectors (Zhang et al., 2024b; Zhang & Pilanci, 2024). However, existing PEFT methods rely on models pre-trained with traditional objectives and do not fully consider their adaptability as universal backbones across diverse tasks.

In contrast, KIND decomposes pre-trained models into learngenes and tailors through knowledge diversion. The class-agnostic knowledge encapsulated in learngenes significantly enhances transfer adaptability, particularly for tasks with large domain shifts compared to the training tasks.

## 3. Methods

### 3.1. Preliminary

#### 3.1.1. LATENT DIFFUSION MODELS

Latent diffusion models transfer the diffusion process from the high-resolution pixel space to the latent space by employing an autoencoder $\mathcal{E}$, which encodes an image $x$ into a latent code $z = \mathcal{E}(x)$. A diffusion model is then trained to generate the corresponding latent code in a denoising process, minimizing the following objective:

$$\mathcal{L} = \mathbb{E}_{z,c,\varepsilon,t}[||\varepsilon - \varepsilon_\theta(z_t|c,t)||_2^2] \qquad (1)$$

Here, $\varepsilon_\theta$ is a noise prediction network that predicts the noise $\varepsilon$ added to $z_t$ at timestep $t$, conditioned on $c$.

#### 3.1.2. DIFFUSION TRANSFORMERS (DITS)

DiT is a transformer-based architecture for noise prediction, replacing the traditional UNet. Given an image $x \in \mathbb{R}^{H_1 \times H_2 \times C}$ and its latent code $z \in \mathbb{R}^{h_1 \times h_2 \times c}$ encoded by $\mathcal{E}$, DiT divides the latent code $z$ into $T$ patches, which are then mapped to $D$-dimensional patch embeddings, with added position embeddings.

The structure of DiTs resembles that of Vision Transformers (ViTs), which comprises $L$ stacked layers, each containing a Multi-Head Self-Attention (MSA) mechanism and a Pointwise Feedforward (PFF) layer. In each layer, a self-attention head $A_i$ performs self-attention using a query $Q$, key $K$,

and value $V \in \mathbb{R}^{T \times D}$, with parameter matrices $W_q^i$, $W_k^i$, and $W_v^i \in \mathbb{R}^{D \times d}$:

$$A_i = \text{softmax}(\frac{Q_i K_i^\top}{\sqrt{d}})V_i , \ A_i \in \mathbb{R}^{T \times d} \qquad (2)$$

MSA mechanism combines $h$ self-attention heads $A$ and projects the concatenated outputs using a weight matrix $W_o$:

$$\text{MSA} = \text{concat}(A_1, A_2, ..., A_h)W_o , \ W_o \in \mathbb{R}^{hd \times D} \qquad (3)$$

In the implementation of MSA, the matrices $W_q^i$, $W_k^i$, and $W_v^i \in \mathbb{R}^{D \times d}$ for $h$ attention heads are combined into three parameter matrices $W_q$, $W_k$, and $W_v \in \mathbb{R}^{D \times hd}$.

PFF layer comprises two linear transformations $W_{in} \in \mathbb{R}^{D \times D'}$ and $W_{out} \in \mathbb{R}^{D' \times D}$ with a GELU (Hendrycks & Gimpel, 2016) activation function:

$$\text{PFF}(x) = \text{GELU}(xW_{in} + b_1)W_{out} + b_2 \qquad (4)$$

where $b_1$ and $b_2$ are the biases for the linear transformations, and $D'$ denotes the hidden layer dimensions.

### 3.2. Knowledge Integration in Weight Matrices

FSGAN (Robb et al., 2020) directly applies SVD to pre-trained model parameters and fine-tunes the singular values for adaptation, achieving success in image segmentation (Sun et al., 2022) and generation (Han et al., 2023) This shows that SVD can create a compact parameter space, facilitating efficient fine-tuning of pre-trained models.

However, directly applying SVD to pre-trained parameter matrices decomposes them based on fixed orthogonalization rules, leading to poor interpretability and making it challenging to determine whether the knowledge in each basic component is class-specific. This limits the model's decomposability, risking the loss of valuable knowledge.

To address this, we integrate knowledge by reconstructing weight matrices using the SVD-derived components $U$, $\Sigma$, and $V$, where each basic component is a combination of a column vector, singular value and row vector from $U$, $\Sigma$, and $V^\top$. We then explicitly associate each basic component with a specific type of knowledge (either class-specific or class-agnostic), which is achieved through a class gate mechanism to divert knowledge (Section 3.3).

For the DiT architecture, the main weight matrices across the $L$-layers are $\theta = \{W_q^{(1\sim L)},\ W_k^{(1\sim L)},\ W_v^{(1\sim L)},\ W_o^{(1\sim L)},\ W_{in}^{(1\sim L)},\ W_{out}^{(1\sim L)}\}$[1]. Let $W_\star^{(l)}$ represent any weight matrix in layer $l$, where $\star \in \mathcal{S}$ and $\mathcal{S} = \{q, k, v, o, in, out\}$ denotes the set of subscripts. The matrices $U_\star^{(l)}$,

---

[1] $W_q^{(1\sim L)}$ denotes the set $\{W_q^{(1)}, W_q^{(2)}, \ldots, W_q^{(L)}\}$. Similar notations throughout the paper follow this convention.

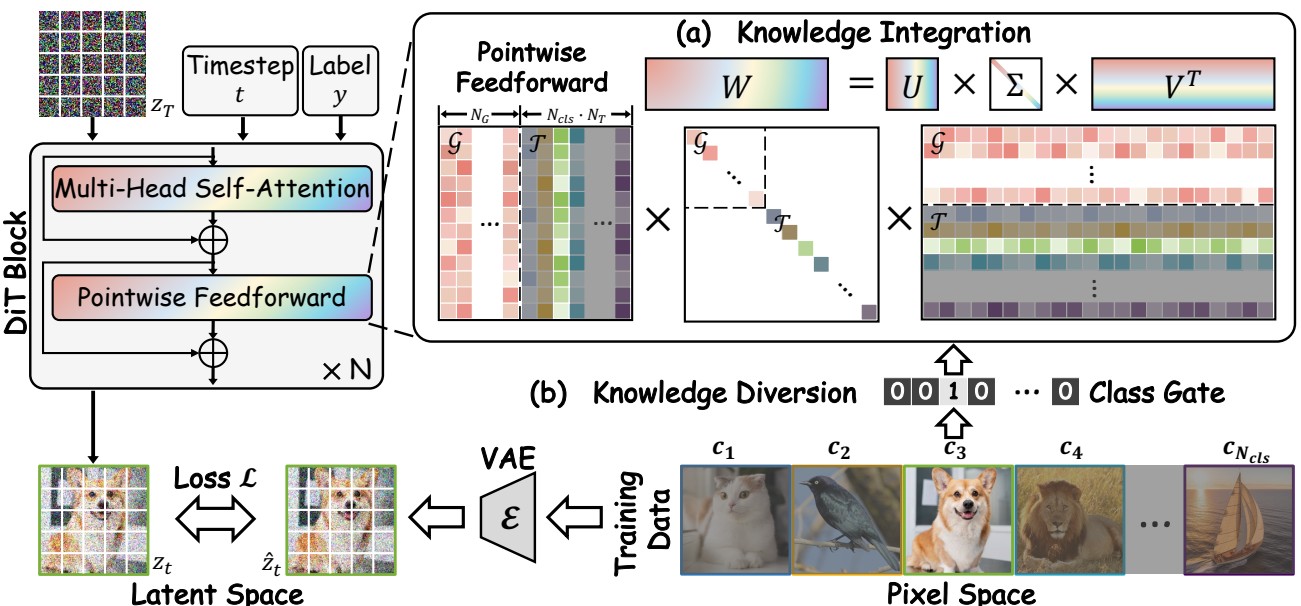

*Figure 2.* (a) For each weight matrix in DiTs, we integrate it into the product of matrices $U$, $\Sigma$ and $V^{\top}$, formally inspired by SVD. The components of these matrices are then explicitly partitioned into the learngenes and tailors, which encapsulate class-agnostic and class-specific knowledge, respectively. (b) Knowledge is diverted through a class gate ensuring each training image updates only the learngenes and their corresponding class-related tailors, so that the class-agnostic knowledge can be condensed into the learngenes, while knowledge specific to each class is diverted into corresponding tailors.

$\Sigma_{\star}^{(l)}$, $V_{\star}^{(l)}$ are the corresponding components that constitute $W_{\star}^{(l)}$, which is calculated as:

$$W_{\star}^{(l)} = U_{\star}^{(l)} \Sigma_{\star}^{(l)} V_{\star}^{(l)^{\top}}$$
$$= \sum_{i=1}^{r} u_{\star}^{(l,i)} \sigma_{\star}^{(l,i)} v_{\star}^{(l,i)} \quad (5)$$

where $\Sigma_{\star}^{(l)} = \mathrm{diag}(\boldsymbol{\sigma})$ with $\boldsymbol{\sigma} = [\sigma_{\star}^{(l,1)}, \sigma_{\star}^{(l,2)}, ..., \sigma_{\star}^{(l,r)}]$. $U_{\star}^{(l)} = [u_{\star}^{(l,1)}, u_{\star}^{(l,2)}, ..., u_{\star}^{(l,r)}] \in \mathbb{R}^{m_1 \times r}$, and $V_{\star}^{(l)} = [v_{\star}^{(l,1)}, v_{\star}^{(l,2)}, ..., v_{\star}^{(l,r)}]^{\top} \in \mathbb{R}^{r \times m_2}$. The rank $r$ and dimensions $m_1$ and $m_2$ are associated with $W_{\star}^{(l)}$. Each basic component is represented as $\Theta_{\star}^{(l,i)} = (u_{\star}^{(l,i)}, \sigma_{\star}^{(l,i)}, v_{\star}^{(l,i)})$.

### 3.3. Knowledge Diversion by Class Labels

Given a dataset with $N_{cls}$ classes, our objective is to allocate knowledge of each class to the corresponding basic components while extracting class-agnostic knowledge shared across all classes, thereby achieving knowledge diversion.

We categorize all basic components into ***learngenes*** and ***tailors***, encapsulating class-agnostic and class-specific knowledge, respectively. Specifically, the components are partitioned based on the number of classes $N_{cls}$ and matrix rank $r$, satisfying $r = N_{cls} \cdot N_T + N_G$, where $N_T$ denotes the number of components per class, with the tailor for the $c$-th class $\mathcal{T}_c$:

$$\mathcal{T}_c = \{\Theta_{\star}^{(l,i)} | i \in [(c-1) \cdot N_T, c \cdot N_T], \star \in \mathcal{S}, l \in [1, L]\} \quad (6)$$

$N_G$ is the number of basic components forming learngenes:

$$\mathcal{G} = \{\Theta_{\star}^{(l,i)} | i \in [N_{cls} \cdot N_T, N_{cls} \cdot N_T + N_G], \star \in \mathcal{S}, l \in [1, L]\} \quad (7)$$

In this way, the $r$ basic components of each matrix are partitioned into $N_G$ learngenes and $N_{cls}$ tailors, with the model parameters represented as $\theta = \mathcal{G} + \sum_{c=1}^{N_{cls}} \mathcal{T}_c$.

To encapsulate the class-specific knowledge of the $c$-th class in the $c$-th tailor, we introduce a class gate $G = [0, \ldots, 0, 1, 0, \ldots, 0] \in \mathbb{R}^{N_{cls}}$ for knowledge diversion during the training of DiTs, where only one the element at the $c$-th position is set to 1, corresponding to the class index. This mechanism ensures that, for each training class, only the weight parameters of the learngene and relevant tailors are updated (See Algorithm 1 for more details). The optimization objective is defined as:

$$\arg\min_{\mathcal{G}, \mathcal{T}} \mathcal{L}_{G \cdot \theta}, \quad \text{s.t. } \theta = \mathcal{G} + \sum_{c=1}^{N_{cls}} \mathcal{T}_c \quad (8)$$

where the loss function $\mathcal{L}$ is defined in Eq. (1).

### 3.4. Decomposable Models for Diverse Scenarios

After training via knowledge diversion, we obtain a decomposable model made up of basic components, which can be adaptively reassembled to meet the target memory size and specific task requirements during deployment.

**Recombination for Variable Model Sizes.** In practice, not all knowledge in pre-trained models is applicable to

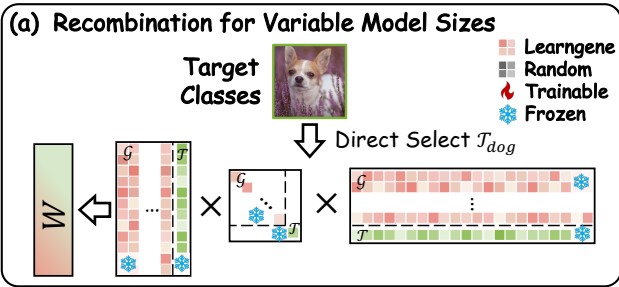

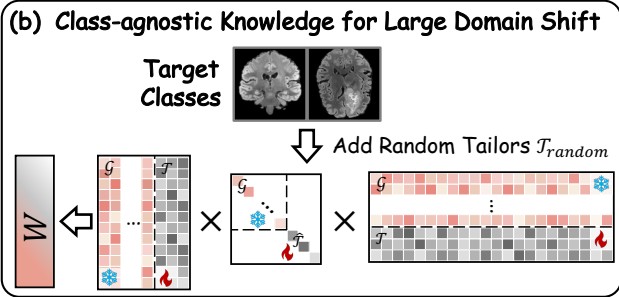

*Figure 3.* (a) For downstream tasks with pre-trained classes, it can directly select the tailors corresponding to the target classes while discarding unrelated ones. (b) When encountering tasks with large domain shifts, only the learngene is transferred, combined with randomly initialized tailors for class-specific fine-tuning.

downstream tasks, and transferring excessive knowledge can be both memory-intensive and redundant. For downstream tasks similar to parts of the training dataset, we can directly select the appropriate pre-trained tailors combined with learngenes. For instance, when deploying a DiT pre-trained on **ImageNet** to a resource-constrained device for generating images of **"dogs"**, we can deploy only the tailor corresponding to "dog" ($\mathcal{T}_{\text{dog}}$) and the learngene ($\mathcal{G}$). Similarly, for unknown classes, we can select closely related tailors for fine-tuning, adjusting the number of tailors based on the available memory.

**Class-agnostic Knowledge for Large Domain Shift.** Pre-trained models often encounter negative transfer when facing large domain shifts, a challenge that also affects the transfer of pre-trained tailors. In such cases, class-agnostic knowledge encapsulated in learngenes fully demonstrates its advantages. Thus, for tasks with large domain shifts, only learngenes need to be transferred, along with randomly initialized tailors $\mathcal{T}_{\text{random}}$. During fine-tuning, we freeze the learngene and only update the tailors, enabling them to learn class-specific knowledge from the downstream task, thereby achieving more efficient fine-tuning.

## 4. Experiments

### 4.1. Datasets

We conduct class-conditioned generation on ImageNet-1K (Deng et al., 2009), which contains 1,000 classes. To

*Table 1.* Performance of constructing variable-sized models on training classes. "Para." denotes the total number of model parameters, which reflects the model size. "Time" is the additional training steps required to construct models of the target sizes.

| | Para.(M) | Methods | Time | FID↓ | sFID↓ | IS↑ | Prec.↑ | Rec.↑ |
|---|---|---|---|---|---|---|---|---|
| **DiT-L** | 457.0 | Trad. PT | 0 | 9.68 | **6.15** | 72.22 | 0.69 | **0.47** |
| | 362.5 | Heur-LG | 100K | 23.86 | 7.24 | 48.34 | 0.54 | 0.47 |
| | 249.2 | Laptop-diff | 100K | 17.20 | 7.25 | 57.07 | 0.59 | 0.47 |
| | 249.2 | Auto-LG | 100K | 18.38 | 8.22 | 57.68 | 0.58 | 0.46 |
| | **245.9** | KIND | **0** | 9.33 | 6.80 | **79.39** | **0.69** | 0.46 |
| **DiT-B** | 129.7 | Trad. PT | 0 | 25.14 | **7.57** | 47.15 | 0.53 | 0.46 |
| | 108.4 | Heur-LG | 100K | 41.53 | 8.93 | 34.29 | 0.42 | 0.47 |
| | 76.5 | Laptop-diff | 100K | 48.22 | 11.09 | 31.19 | 0.37 | **0.47** |
| | 76.5 | Auto-LG | 100K | 45.69 | 10.77 | 32.77 | 0.39 | 0.47 |
| | **70.2** | KIND | **0** | **21.14** | 8.85 | **58.18** | **0.55** | 0.44 |

minimize inter-class similarity, we merge certain similar classes based on their superclasses in WordNet (Miller, 1995), resulting in a final set of 611 classes. Among these, 150 classes are used for pre-training the diffusion models, while the remaining 461 classes serve as novel classes for constructing downstream tasks. Further details can be found in Appendix A.3. Additionally, we use datasets, including CelebA-HQ (Huang et al., 2018), Hubble (Weinzierl, 2023), MRI, and Pokémon, to simulate large domain shifts compared to the training data.

### 4.2. Basic Setting

For pre-training DiT, we train class-conditional latent DiTs of sizes -B and -L, with a latent patch size of $p = 2$ at a $256 \times 256$ image resolution on training classes. All models are trained using AdamW with a batch size of 256 and a constant learning rate of $1 \times 10^{-4}$ over 300K steps. An exponential moving average (EMA) of DiT weights is used with a decay rate of 0.9999, and results are reported using the EMA model. During image generation, a classifier-free guidance (cfg) scale of 1.5 is applied. Performance is evaluated using Fréchet Inception Distance (FID) (Heusel et al., 2017), sFID (Nash et al., 2021), Fréchet DINO distance(FDD) (Stein et al., 2023), Inception Score (Salimans et al., 2016) and Precision/Recall (Kynkäänniemi et al., 2019). Further details are provided in Appendix A.2.

## 5. Results

### 5.1. Construction of Variable-Sized Pre-Trained Models

The models pre-trained by KIND are inherently decomposable, consisting of *learngenes* that encapsulate class-agnostic knowledge and *tailors* that capture class-specific knowledge. This decomposition enables flexible deployment of models across devices, as demonstrated in Table 1.

Compared to traditional pre-trained models, KIND achieves comparable performance with the same number of training

*Table 2.* Performance of various PEFT and learngene methods on novel classes. All methods are fine-tuned for 50K steps on 18 downstream tasks involving novel classes. "Para." denotes the average number of trainable parameters, while "FLOPs" represents the average total floating-point operations required during fine-tuning.

| Methods | | DiT-B/2 | | | | | | DiT-L/2 | | | | | |
|---|---|---|---|---|---|---|---|---|---|---|---|---|---|
| | | Para.(M) | FLOPs(G) | FID↓ | sFID↓ | IS↑ | Prec.↑ | Recall↑ | Para.(M) | FLOPs(G) | FID↓ | sFID↓ | IS↑ | Prec.↑ | Recall↑ |
| PEFT | SVDiff | *0.1* | *43.6* | 55.01 | 18.12 | 19.6 | 0.35 | 0.55 | *0.2* | *155.0* | 49.59 | 16.81 | 20.8 | 0.38 | 0.56 |
| | OFT | *14.2* | *119.7* | 36.19 | 17.79 | 32.0 | 0.48 | 0.50 | *50.5* | *425.6* | 24.81 | 18.27 | 44.1 | 0.59 | 0.47 |
| | LoRA | *12.8* | *50.1* | 36.70 | 16.28 | 31.6 | 0.44 | 0.57 | *45.3* | *178.2* | 22.55 | 14.00 | 46.3 | 0.55 | 0.56 |
| | PiSSA | *12.8* | *50.1* | 33.16 | 15.51 | 34.6 | 0.49 | 0.52 | *45.3* | *178.2* | 19.41 | 14.72 | 53.7 | 0.63 | 0.50 |
| | LoHa | *12.7* | *87.1* | 42.38 | 17.37 | 27.3 | 0.40 | **0.58** | *45.3* | *309.6* | 29.79 | 15.17 | 35.8 | 0.49 | **0.59** |
| | DoRA | *12.8* | *129.5* | 35.87 | 16.40 | 32.3 | 0.45 | 0.56 | *45.6* | *503.0* | 21.28 | 14.16 | 48.3 | 0.57 | 0.55 |
| LG | Heur-LG | *129.6* | *43.6* | 55.45 | 22.14 | 24.4 | 0.33 | 0.48 | *456.8* | *155.0* | 41.83 | 19.23 | 30.9 | 0.40 | 0.51 |
| | Auto-LG | *129.6* | *43.6* | 56.38 | 21.39 | 25.5 | 0.30 | 0.49 | *456.8* | *155.0* | 31.78 | 18.71 | 41.7 | 0.46 | 0.54 |
| | KIND | *12.8* | *33.7* | **20.94** | **14.75** | **62.4** | **0.53** | 0.50 | *45.4* | *119.6* | **12.87** | **12.93** | **86.1** | **0.65** | 0.51 |
| FT | Full FT | *129.6* | *43.6* | 26.49 | 15.08 | 45.1 | 0.51 | 0.55 | *456.8* | *155.0* | 14.51 | 13.16 | 69.1 | 0.63 | 0.55 |

steps, without increasing training complexity. Additionally, the decomposable nature of KIND allows for direct recombination tailored to specific deployment needs, with ***no further time-consuming*** steps required. In contrast to knowledge distillation and pruning (Zhang et al., 2024a), KIND offers significant advantages by avoiding the resource overhead of ***repeated distillation and pruning*** for each model size, which is required in distillation-based methods.

Unlike traditional learngenes, such as Heur-LG (Wang et al., 2022) and Auto-LG (Wang et al., 2023), which directly transfer certain layers from traditional pre-trained models, KIND encapsulates task-agnostic knowledge into learngenes and retains task-specific knowledge in tailors through knowledge diversion. This enables the direct combination of learngenes and tailors without additional training, ensuring both efficiency and adaptability across tasks.

### 5.2. Performance on Tasks with Novel Classes

To evaluate KIND's adaptability, we use learngenes as the backbone with randomly initialized tailors and compare it to PEFT methods based on traditional pre-trained models on tasks with novel classes. As shown in Table 2, KIND achieves state-of-the-art results on DiT-B and DiT-L, reducing FID by 6.54 and sFID by 1.07, while using only 45.4M parameters and saving 35.4G FLOPs on DiT-L.

Despite the efficiency of PEFT methods, a significant performance gap remains compared to Full FT, highlighting the task discrepancy between training and novel classes. PEFT methods, which freeze pre-trained parameters, struggle to adapt to novel tasks. As shown in Figure 4, PEFT-generated images perform poorly in capturing class-specific knowledge due to limited trainable parameters and task mismatch. Existing learngene methods like Heur-LG and Auto-LG transfer partial knowledge from pre-trained models, but the transferability of each module, trained with traditional objectives, is limited.

*Table 3.* Performance comparison of KIND and PEFT methods in transferring to downstream tasks with significant domain shifts, evaluated using FDD for image quality assessment.

| | CelebA-HQ | | Hubble | | MRI | | Pokemon | |
|---|---|---|---|---|---|---|---|---|
| | DiT-B | DiT-L | DiT-B | DiT-L | DiT-B | DiT-L | DiT-B | DiT-L |
| SVDiff | 0.622 | 0.388 | 0.385 | 0.305 | 0.187 | 0.148 | 0.605 | 0.469 |
| OFT | 0.343 | 0.226 | 0.255 | 0.168 | 0.056 | 0.046 | 0.469 | 0.321 |
| LoRA | 0.284 | 0.197 | 0.232 | 0.142 | 0.061 | 0.056 | 0.412 | 0.285 |
| PiSSA | 0.281 | 0.195 | 0.211 | 0.152 | 0.057 | 0.051 | 0.418 | 0.295 |
| LoHa | 0.336 | 0.268 | 0.252 | 0.189 | 0.065 | 0.130 | 0.439 | 0.316 |
| DoRA | 0.282 | 0.203 | 0.589 | 0.330 | 0.043 | 0.048 | 0.396 | 0.333 |
| KIND | **0.201** | **0.152** | **0.124** | **0.109** | **0.042** | **0.040** | **0.343** | **0.262** |

In contrast, KIND diverts class-agnostic knowledge into learngenes, creating a flexible backbone for adaptation to downstream tasks with novel classes. The randomly initialized tailors are adjusted via low-rank assumptions, combining with learngenes to meet task-specific needs, thereby improving transfer efficiency and enhancing the generalizability of knowledge transfer. As shown in Figure 4 and Table 2, KIND-generated images outperform PEFT methods in both quality and performance metrics.

### 5.3. Performance on Tasks with Large Domain Shifts

KIND demonstrates significant advantages in adapting to tasks with novel classes, with these benefits becoming even more pronounced when dealing with tasks involving large domain shifts. As shown in Table 3 and Figure 5, KIND outperforms PEFT methods on both DiT-B and DiT-L, achieving substantial improvements in image generation quality.

This further demonstrates that the knowledge encapsulated in learngenes is sufficiently class-agnostic, allowing it to be shared effectively across various tasks. In contrast, PEFT methods based on traditional pre-trained models show disadvantages, as the knowledge learned from ImageNet is often difficult to transfer to new domains, especially in specialized fields like Hubble and MRI. This highlights a key limitation

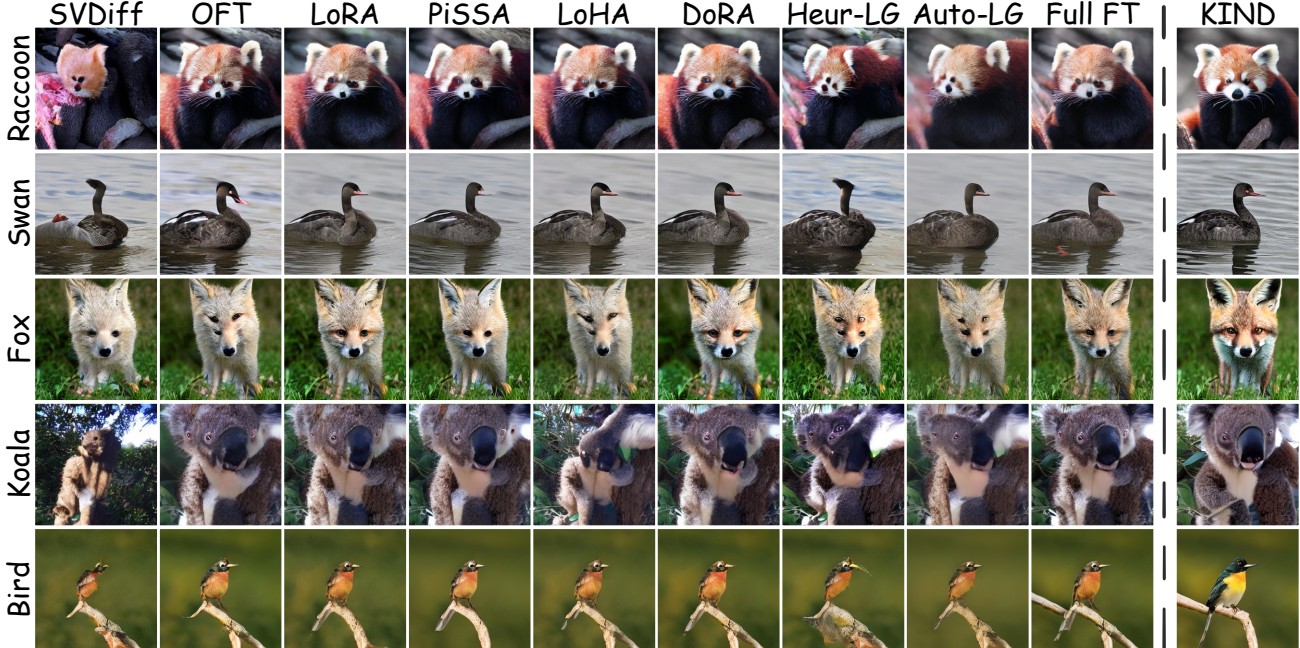

*Figure 4.* Selected samples from tasks with novel classes, generated by KIND and other PEFT methods using the DiT-L/2 model, with a resolution of $256 \times 256$. All images are generated using a classifier-free guidance (cfg) scale of 3.0.

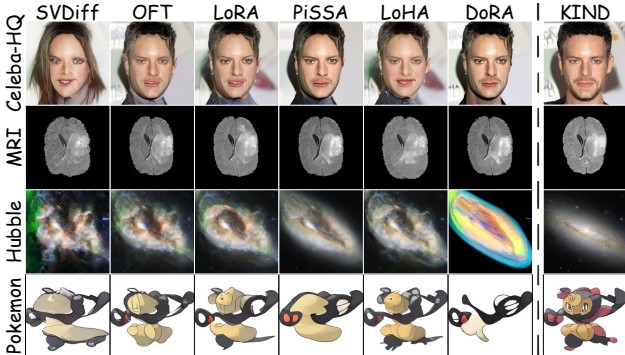

*Figure 5.* Selected samples from tasks with large domain shifts, generated by KIND and other PEFT methods using the DiT-L/2, with a resolution of $256 \times 256$. All images are generated using a classifier-free guidance (cfg) scale of 1.5.

of current pre-training approaches, which aim to improve generalization by incorporating as many domain-specific images as possible during training (Ramesh et al., 2022; Esser et al., 2024). While this may enhance performance, it leads to larger model sizes, reduced transfer flexibility, and increased computational overhead.

### 5.4. Ablation and Analysis

#### 5.4.1. ABLATION EXPERIMENTS

To assess the effectiveness of learngenes, tailors, and the class gate, we conduct a series of ablation experiments. #1 performs Singular Value Decomposition (SVD) on pre-

*Table 4.* Ablation study on different components of KIND.

| | | LG | Tailor | Gate | FID↓ | sFID↓ | IS↑ | Prec.↑ | Recall↑ |
|---|---|---|---|---|---|---|---|---|---|
| **DiT-B/2** | #1 | | | | 60.28 | 19.96 | 20.4 | 0.30 | 0.49 |
| | #2 | ✓ | | | 49.54 | 18.08 | 23.2 | 0.34 | **0.56** |
| | #3 | ✓ | ✓ | | 21.60 | 14.84 | 59.7 | **0.54** | 0.50 |
| | KIND | ✓ | ✓ | ✓ | **20.94** | **14.75** | **62.4** | 0.53 | 0.50 |
| **DiT-L/2** | #1 | | | | 42.04 | 18.07 | 28.0 | 0.41 | 0.54 |
| | #2 | ✓ | | | 33.53 | 15.55 | 32.2 | 0.46 | **0.59** |
| | #3 | ✓ | ✓ | | 13.03 | 12.93 | 85.1 | 0.64 | 0.51 |
| | KIND | ✓ | ✓ | ✓ | **12.87** | 12.93 | **86.1** | **0.65** | 0.51 |

trained weights and randomly selects $N_G$ singular vectors to form its backbone, followed by fine-tuning with LoRA. #2 replaces the backbone with learngenes extracted by KIND, based on the structure in #1. #3 substitutes tailors for LoRA in fine-tuning the model, without using the class gate.

As shown in Table 4, the knowledge encapsulated in learngenes, which undergoes knowledge diversion, is more class-agnostic, making it better suited for adaptation to downstream tasks, especially when these tasks differ significantly from the training tasks (e.g., #1 vs. #2). Additionally, tailors can also function as a PEFT method by integrating class-specific knowledge into pre-trained models or learngenes, thereby enhancing the model's ability to acquire new knowledge for downstream tasks (#2 vs. #3). Finally, the class gate further enhances this by helping the model distinguish class-specific knowledge, boosting the effectiveness of the tailors (#3 vs. KIND).

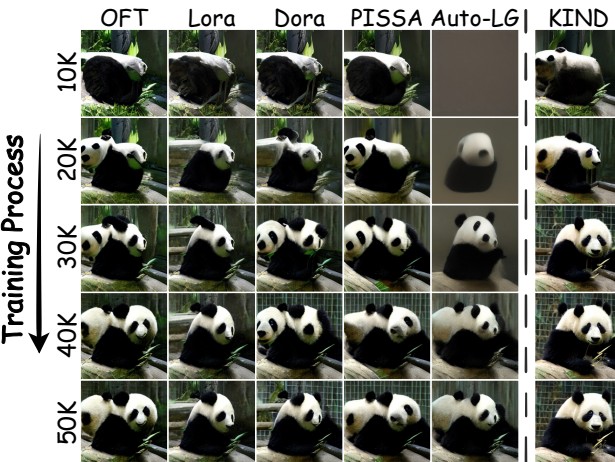

*Figure 6.* Visualization of convergence speed of KIND and other methods on downstream tasks. Each image is sampled every 10K steps to illustrate progress more clearly.

*Table 5.* Comparison of pre-trained models and learngenes when serving as backbones on training tasks.

|  | Entropy↑ | Variance↓ | Kurtosis↓ |
|---|---|---|---|
| Raw Images of ImageNet | 1.458 | 6.414e$^{-4}$ | 884.3 |
| Pretrained Model | 2.387 | 4.516e$^{-4}$ | 780.1 |
| Learngene | **4.046** | **1.495e$^{-4}$** | **544.9** |

### 5.4.2. STRONG LEARNING ABILITY BROUGHT BY LEARNGENES

As noted in (Wang et al., 2022; Xia et al., 2024), learngenes accelerate downstream model adaptation by transferring common knowledge, offering a significant advantage over training from scratch. Beyond this, KIND further improves convergence speed compared to PEFT methods. Figure 6 illustrates the convergence speed of KIND, with images generated by models every 10K training steps.

The convergence speed is generally influenced by the number of trainable parameters during fine-tuning, with PEFT methods focusing on reducing this number using techniques like orthogonalization and low-rank constraints (Ding et al., 2023; Han et al., 2024). However, these methods often neglect the transferability of knowledge in pre-trained models by directly fixing their parameters. In contrast, KIND leverages learngenes that encapsulate class-agnostic knowledge as the backbone, offering superior transferability while remaining lightweight. Meanwhile, the tailors capture task-specific knowledge, allowing KIND to achieve faster convergence and improved performance on downstream tasks.

### 5.4.3. ANALYSIS ON CLASS-AGNOSTIC KNOWLEDGE

As discussed earlier, learngenes provide a superior backbone compared to pre-trained models by encapsulating class-

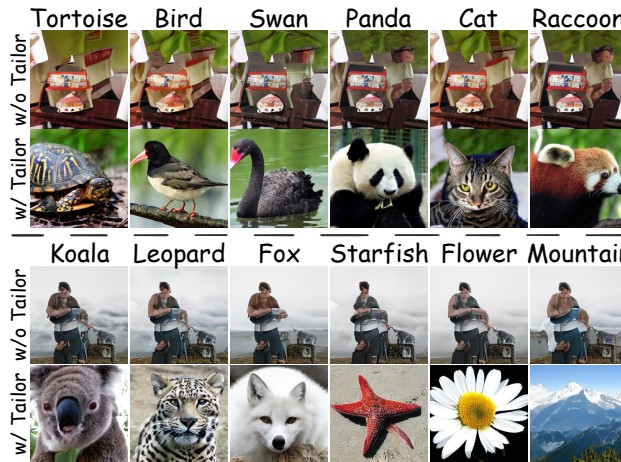

*Figure 7.* Visualization of KIND w/ and w/o Tailers (i.e., learngene only) across 12 superclasses for 2 different seeds.

agnostic knowledge. To further investigate this, we analyze the properties of the class-agnostic knowledge encapsulated in learngenes. Table 5 compares learngenes themselves (i.e., w/o tailors) with pre-trained models on training tasks. The results reveal that learngenes demonstrate higher entropy, along with lower variance and kurtosis, suggesting that the class-agnostic knowledge they encapsulate is widely applicable across diverse classes. Such stability underscores that learngenes, as a backbone, offer better adaptability to unfamiliar classes than traditional pre-trained models.

We also visualize learngenes with and without tailors in Figure 7. The visualizations demonstrate that learngenes are not sensitive to category variations, consistently generating similar images across different class conditions. While these images may lack detailed semantic information on their own, combining them with class-specific knowledge (i.e., tailors) enables the generation of images corresponding to specific classes. This further underscores the inherent commonality of knowledge within learngenes.

## 6. Conclusion

In this study, we introduce KIND, a pre-training method for constructing decomposable models. KIND employs knowledge diversion during pre-training, separating class-agnostic knowledge into learngenes and class-specific knowledge into tailors. This approach enables the adaptive assembly of variable-sized models by selectively integrating relevant tailors. The class-agnostic knowledge within learngenes mitigates the challenges of tasks with large domain shifts, particularly when combined with randomly initialized tailors for task-specific fine-tuning. We demonstrate the effectiveness of KIND in resource-constrained scenarios and tasks with significant domain shifts, with further analysis and visualizations illustrating the robustness of the class-agnostic knowledge encapsulated in learngenes.

## Acknowledgement

We sincerely appreciate Freepik for contributing to the figure design. This research was supported by the Jiangsu Science Foundation (BK20243012, BG2024036, BK20230832), the National Science Foundation of China (62125602, U24A20324, 92464301, 62306073), China Postdoctoral Science Foundation (2022M720028), and the Xplorer Prize.

## Impact Statement

The broader impact of our work lies in how KIND redefines the training objectives of pre-trained models, enabling the construction of decomposable models that can be recombined to create models with variable sizes. This approach facilitates faster deployment, reduces resource consumption, and enhances adaptability across various tasks and datasets, offering significant value for both research and industrial applications in AI model scaling and transfer learning.

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

# A. Training Details

## A.1. Details of Knowledge Diversion

Algorithm 1 presents the pseudo code for diverting class-agnostic knowledge into learngenes and class-specific knowledge into tailors.

---

**Algorithm 1** Diversion of Class-agnostic Knowledge and Class-specific Knowledge

---

**Input**: DiT $f$, Training dataset $\mathcal{D} = \{(x^{(i)}, y^{(i)})\}_{i=1}^{m}$ of $N_{cls}$ classes, number of epochs $N_{ep}$, batch size $B$, learning rate $\alpha$
**Output**: Learngene $\mathcal{G}$

1: Randomly initialize the weight matrices $\theta$ of $f$, as well as the matrices $U_\star^{(l)}$, $\Sigma_\star^{(l)}$, and $V_\star^{(l)}$
2: **for** $ep = 1$ to $N_{ep}$ **do**
3:    **for** each batch $\{(x_i, y_i)\}_{i=1}^{B}$ **do**
4:       Update $\theta$ of $f$ with $U_\star^{(l)}$, $\Sigma_\star^{(l)}$ and $V_\star^{(l)}$ under the rule of Eq. (5)
5:       Initialize class gate $G \in \mathbb{R}^{B \times N_{cls}}$ according to labels of images in this batch
6:       For each $x_i$, forward propagate $\hat{y}_i = f(x_i, G \cdot \theta)$
7:       Calculate $\mathcal{L}_{\text{batch}} = \frac{1}{B}\sum_{i=1}^{B} \mathcal{L}(\hat{y}_i, y_i)$ according to Eq. (1)
8:       Backward propagate the loss $\mathcal{L}_{\text{batch}}$ to compute the gradients with respect to $U_\star^{(l)}$, $\Sigma_\star^{(l)}$ and $V_\star^{(l)}$: $\nabla_U \mathcal{L}_{\text{batch}}, \nabla_\Sigma \mathcal{L}_{\text{batch}}$ and $\nabla_V \mathcal{L}_{\text{batch}}$
9:       Update the learngenes $U_{G,\star}^{(l)}$, $\Sigma_{G,\star}^{(l)}$ and $V_{G,\star}^{(l)}$:
$$U_{G,\star}^{(l)} := U_{G,\star}^{(l)} - \alpha \cdot \nabla_U \mathcal{L}_{\text{batch}},$$
$$\Sigma_{G,\star}^{(l)} := \Sigma_{G,\star}^{(l)} - \alpha \cdot \nabla_\Sigma \mathcal{L}_{\text{batch}}$$
$$V_{G,\star}^{(l)} := V_{G,\star}^{(l)} - \alpha \cdot \nabla_V \mathcal{L}_{\text{batch}}$$
10:     Update the tailors $U_{T_i,\star}^{(l)}$, $\Sigma_{T_i,\star}^{(l)}$ and $V_{T_i,\star}^{(l)}$:
$$U_{T_i,\star}^{(l)} := U_{T_i,\star}^{(l)} - \alpha \cdot G(\nabla_U \mathcal{L}_{\text{batch}})$$
$$\Sigma_{T_i,\star}^{(l)} := \Sigma_{T_i,\star}^{(l)} - \alpha \cdot G(\nabla_\Sigma \mathcal{L}_{\text{batch}})$$
$$V_{T_i,\star}^{(l)} := V_{T_i,\star}^{(l)} - \alpha \cdot G(\nabla_V \mathcal{L}_{\text{batch}})$$
11:    **end for**
12: **end for**

---

## A.2. Hyper-parameters

Table 6 presents the basic settings, including learning rate, training steps and the number of learngene components $N_G$ and tailor components $N_T$ for KIND integrating and diverting knowledge. And Table 7 presents the hyper-parameters of PEFT and other learngene methods on 18 downstream tasks. Apart from general hyper-parameters, we also record the hyper-parameters specific to each method. Among them, the parameter $r$ of Lora, PiSSA, Dora and LoHA denotes the rank and the $r$ in OFT denotes the block number respectively.

## A.3. Details of Downstream Tasks

Table 9 presents the details of 18 downstream tasks, which are sorted by the class numbers in each task. Each task is composed of $c \in [7, 35]$ novel classes, where the classes

*Table 6.* Hyper-parameters for KIND diverting knowledge on training classes of ImageNet-1K.

| Training Settings | Configuration |
|---|---|
| optimizer | AdamW |
| learning rate | 1e-4 |
| weight decay | 0 |
| batch size | 256 |
| training steps | 200,000 |
| image size | 256×256 |
| VAE | ema |
| DiT block | adaLN-Zero |
| $N_G$ (DiT-B/-L) | 318 / 424 |
| $N_T$ (DiT-B/-L) | 3 / 4 |

merged into superclasses in ImageNet1K and their corresponding superclasses are listed in Table 10 and Table 11, while the rest remain the same as the classes in ImageNet-1K.

# B. Additional Results

We provide more images of novel classes generated by our KIND which is a DiT-L/2 model composed of learngenes and tailors at $256 \times 256$ resolution, as shown in Figure 8-15.

*Table 7.* Hyper-parameters for PEFT and learngene methods when fine-tuning on novel classes of ImageNet-1K.

| Methods | Batch Size | Training Steps | Learning Rate (DiT-B / -L) | Task ID | #1 | #2 | #3 | #4 | #5 | #6 | #7 | #8 | #9 | #10 | #11 | #12 | #13 | #14 | #15 | #16 | #17 | #18 |
|---|---|---|---|---|---|---|---|---|---|---|---|---|---|---|---|---|---|---|---|---|---|---|
| | | | | | | | | | | | | | | | | | | | **Rank or Block Number** $r$ | | | | |
| OFT | 256 | 50K | 1e-4 | | 21 | 11 | 8 | 7 | 6 | 6 | 5 | 5 | 5 | 5 | 5 | 5 | 4 | 4 | 4 | 4 | 4 | 4 |
| Lora | 512 | 50K | 1e-3 | **-B** | 21 | 39 | 54 | 60 | 69 | 72 | 78 | 78 | 78 | 84 | 84 | 87 | 90 | 90 | 93 | 99 | 102 | 105 |
| | | | | **-L** | 28 | 52 | 72 | 80 | 92 | 96 | 104 | 104 | 104 | 112 | 112 | 116 | 120 | 120 | 124 | 132 | 136 | 140 |
| PiSSA | 256 | 50K | 1e-3 | **-B** | 21 | 39 | 54 | 60 | 69 | 72 | 78 | 78 | 78 | 84 | 84 | 87 | 90 | 90 | 93 | 99 | 102 | 105 |
| | | | | **-L** | 28 | 52 | 72 | 80 | 92 | 96 | 104 | 104 | 104 | 112 | 112 | 116 | 120 | 120 | 124 | 132 | 136 | 140 |
| Dora | 256 | 50K | 1e-3 | **-B** | 21 | 39 | 54 | 60 | 69 | 72 | 78 | 78 | 78 | 84 | 84 | 87 | 90 | 90 | 93 | 99 | 102 | 105 |
| | | | | **-L** | 28 | 52 | 72 | 80 | 92 | 96 | 104 | 104 | 104 | 112 | 112 | 116 | 120 | 120 | 124 | 132 | 136 | 140 |
| LoHA | 256 | 50K | 1e-3 | **-B** | 10 | 19 | 27 | 30 | 34 | 36 | 39 | 39 | 39 | 42 | 42 | 43 | 45 | 45 | 46 | 49 | 51 | 52 |
| | | | | **-L** | 14 | 26 | 36 | 40 | 46 | 48 | 52 | 52 | 52 | 56 | 56 | 58 | 60 | 60 | 62 | 66 | 68 | 70 |
| SVDiff | 256 | 50K | 5e-3/3e-3 | | | | | | | | | | —— | | | | | | | | | |
| Heru-LG | 256 | 50K | 1e-4 | | | | | | | | | | —— | | | | | | | | | |
| Auto-LG | 256 | 50K | 1e-4 | | | | | | | | | | —— | | | | | | | | | |
| KIND | 256 | 50K | 1e-3 | | | | | | | | | | —— | | | | | | | | | |
| Full FT | 256 | 50K | 1e-4 | | | | | | | | | | —— | | | | | | | | | |

*Table 8.* Detailed FID of PEFT and learngene methods when fine-tuning on each novel classes.

| | Methods | #1 | #2 | #3 | #4 | #5 | #6 | #7 | #8 | #9 | #10 | #11 | #12 | #13 | #14 | #15 | #16 | #17 | #18 |
|---|---|---|---|---|---|---|---|---|---|---|---|---|---|---|---|---|---|---|---|
| | | | | | | | | | | **Task ID** | | | | | | | | | |
| **DiT-B** PEFT | SVDiff | 143.4 | 144.9 | 140.6 | 112.3 | 112.5 | 114.2 | 117.4 | 104.3 | 108.6 | 107.5 | 102.3 | 93.6 | 97.5 | 108.9 | 109.6 | 95.2 | 81.6 | 100.6 |
| | OFT | 92.3 | 90.4 | 93.7 | 71.9 | 76.0 | 86.7 | 82.3 | 72.6 | 74.5 | 76.9 | 65.4 | 63.0 | 67.8 | 77.3 | 78.1 | 64.4 | 63.7 | 75.2 |
| | Lora | 85.5 | 94.2 | 97.7 | 75.8 | 80.3 | 89.2 | 89.4 | 76.6 | 76.2 | 83.1 | 68.9 | 64.7 | 70.5 | 78.7 | 79.1 | 67.0 | 63.3 | 78.6 |
| | PiSSA | 83.0 | 89.4 | 93.0 | 69.3 | 73.8 | 82.1 | 81.1 | 69.4 | 71.6 | 76.2 | 64.3 | 60.5 | 64.3 | 74.4 | 70.2 | 60.7 | 59.7 | 71.1 |
| | LoHa | 94.9 | 100.8 | 108.3 | 84.3 | 88.2 | 95.8 | 97.5 | 85.5 | 86.6 | 90.6 | 78.9 | 73.2 | 79.3 | 88.8 | 88.0 | 76.4 | 69.4 | 86.9 |
| | Dora | 82.9 | 91.6 | 94.0 | 73.1 | 77.8 | 87.2 | 87.8 | 73.9 | 75.4 | 79.0 | 67.8 | 64.2 | 69.6 | 77.0 | 78.6 | 65.2 | 62.2 | 77.0 |
| LG | Heru-LG | 98.7 | 111.1 | 122.4 | 97.0 | 102.5 | 114.4 | 122.2 | 95.1 | 99.5 | 108.4 | 87.8 | 90.5 | 91.7 | 103.6 | 101.8 | 94.4 | 88.3 | 100.2 |
| | Auto-LG | 107.8 | 113.7 | 129.3 | 105.6 | 100.1 | 117.7 | 112.3 | 100.5 | 100.3 | 105.7 | 89.9 | 91.4 | 93.7 | 105.1 | 101.7 | 99.1 | 87.3 | 99.9 |
| | KIND | **55.0** | **73.4** | **70.4** | **52.7** | **58.3** | **65.2** | **59.7** | **47.8** | **51.9** | **56.7** | **42.7** | **43.7** | **44.6** | **56.3** | **62.8** | **43.5** | **39.8** | **52.0** |
| FT | Full FT | 56.3 | 75.5 | 78.1 | 59.9 | 65.1 | 72.6 | 70.1 | 58.6 | 60.1 | 66.1 | 54.2 | 51.4 | 53.8 | 63.7 | 63.3 | 52.8 | 51.5 | 62.7 |
| **DiT-L** PEFT | SVDiff | 118.2 | 132.0 | 127.2 | 98.3 | 97.2 | 103.0 | 105.6 | 92.3 | 98.3 | 97.2 | 92.9 | 84.1 | 90.5 | 102.3 | 110.4 | 109.0 | 76.3 | 92.3 |
| | OFT | 59.4 | 71.4 | 72.4 | 52.3 | 57.9 | 65.1 | 64.5 | 55.1 | 60.4 | 58.6 | 48.7 | 50.1 | 52.7 | 62.2 | 60.8 | 50.1 | 51.1 | 61.9 |
| | Lora | 54.6 | 72.5 | 72.0 | 55.9 | 59.0 | 65.7 | 65.1 | 53.7 | 54.5 | 61.6 | 48.7 | 49.4 | 50.3 | 57.4 | 57.1 | 47.5 | 46.1 | 59.7 |
| | PiSSA | 52.6 | 68.9 | 67.0 | 50.2 | 54.1 | 60.8 | 58.4 | 49.2 | 48.4 | 55.4 | 43.1 | 44.4 | 44.3 | 53.1 | 48.6 | 41.1 | 41.9 | 50.6 |
| | LoHa | 65.3 | 78.7 | 83.3 | 63.9 | 69.6 | 77.6 | 78.3 | 66.1 | 66.7 | 73.8 | 62.2 | 59.0 | 62.5 | 68.6 | 68.1 | 59.3 | 56.1 | 72.5 |
| | Dora | 52.2 | 71.3 | 68.0 | 52.9 | 56.7 | 64.3 | 62.4 | 52.2 | 51.1 | 58.7 | 46.9 | 47.0 | 47.9 | 56.0 | 55.7 | 44.9 | 45.1 | 56.2 |
| LG | Heru-LG | 73.3 | 92.6 | 97.1 | 79.9 | 86.2 | 94.4 | 94.5 | 77.8 | 82.2 | 88.8 | 72.4 | 71.9 | 77.2 | 85.8 | 85.1 | 74.8 | 71.9 | 84.0 |
| | Auto-LG | 66.5 | 81.1 | 82.6 | 69.3 | 70.0 | 80.4 | 76.3 | 66.6 | 67.4 | 72.8 | 58.8 | 59.3 | 61.0 | 70.9 | 69.3 | 64.9 | 58.1 | 70.2 |
| | KIND | 39.0 | 66.2 | 61.8 | 44.2 | 46.0 | **54.7** | **47.5** | 39.1 | 40.0 | 46.3 | 33.2 | 36.3 | 34.7 | 45.9 | 43.9 | **31.5** | **30.9** | 40.8 |
| FT | Full FT | **38.0** | **64.1** | 61.9 | 44.6 | **45.5** | 56.1 | 50.8 | 41.4 | 41.2 | 48.5 | 36.1 | 38.6 | 38.2 | 47.4 | **43.4** | 35.3 | 34.9 | 44.1 |

*Table 9.* Details of superclasses in each downstream task

| Task | Superclasses of ImageNet |
|------|--------------------------|
| **#1** | n02510455 n02509815 n01662784 n02118333 n02083346 n02437616 n02457408 |
| **#2** | n03187595 n03788365 n03933933 n04273569 n03843555 n03400231 n03325584 n09472597 n03874293 n04591713 n03854065 n03868863 n07711569 |
| **#3** | n07753592 n03763968 n03109150 n09399592 n03903868 n03720891 n02939185 n03908714 n04014297 n02804414 n06785654 n04131690 n02794156 n02971356 n02056570 n02965783 n04243546 n06359193 |
| **#4** | n02877765 n04238763 n04009552 n03666591 n07614500 n09332890 n01629276 n04483307 n03291819 n02120997 n03717622 n04041544 n03873416 n04467665 n03394916 n03272010 n04118538 n04367480 n04447861 n03775071 |
| **#5** | n04086273 n04141076 n03657121 n03379051 n02401031 n01503061 n03840681 n04380533 n03871628 n11879895 n04090263 n04557648 n03016953 n02808304 n02879718 n03724870 n04423845 n02917067 n03691459 n02672831 n04146614 n04525305 n04264628 |
| **#6** | n03496892 n06874185 n04392985 n03485794 n03982430 n04540053 n03602883 n02871525 n02978881 n03961711 n04005630 n03065424 n04200800 n02823750 n03344393 n04325704 n03220513 n03498962 n04356056 n03347037 n09421951 n07760859 n04133789 n07565083 |
| **#7** | n04332243 n02883205 n03405725 n03017168 n04553703 n03777568 n02951358 n07720875 n03637318 n02090827 n04265275 n03028079 n07920052 n03954731 n04141327 n03255030 n03447447 n00002684 n03530642 n03425413 n04524313 n03110669 n03764736 n12267677 n02676566 n03417042 |
| **#8** | n03676483 n02865351 n03792972 n02974003 n02906734 n07860988 n03249569 n00021265 n02727426 n03782006 n02317335 n02815834 n03388043 n03529860 n02817516 n03761084 n09246464 n03899768 n03970156 n04485082 n01769347 n07880968 n03197337 n03876231 n02699494 n03472232 |
| **#9** | n02121808 n07734744 n03424325 n03494278 n03935335 n03690938 n03240683 n03467068 n02980441 n03450230 n02512053 n04517823 n02730930 n03133878 n03259280 n04376876 n03803284 n03920288 n02966193 n02814860 n02669723 n03000134 n02793495 n02766320 n03649909 n04125021 |
| **#10** | n03985232 n03590841 n03388549 n04065272 n03633091 n02916936 n03201208 n04208210 n02988304 n09229709 n02769748 n02791270 n03814639 n03481172 n03692522 n04501370 n03584829 n02843684 n04252225 n03196217 n02704792 n03384352 n03785016 n03459775 n03599486 n01806143 n03294048 n03995372 |
| **#11** | n04341686 n03603722 n04081281 n03623198 n03497657 n02690373 n09193705 n04486054 n01986214 n01639765 n03180011 n03532672 n03540267 n02356798 n03662601 n04277352 n04204238 n04204347 n04530566 n04033901 n03793489 n02268148 n04209239 n04266014 n01861778 n03062245 n03179701 n11939491 |
| **#12** | n04111531 n04597913 n07932039 n04118776 n02859443 n04523525 n02077923 n03938244 n07707451 n04371430 n02797295 n04228054 n03207743 n01882714 n07716906 n03216828 n04589890 n03063689 n03630383 n04252077 n02153203 n03207941 n03908618 n03796401 n07697313 n02898711 n04548362 n03290653 n02930766 |
| **#13** | n03000247 n04040759 n04590129 n03492542 n03733805 n04044716 n01877812 n04418357 n09428293 n03045698 n03998194 n03443371 n03983396 n03902125 n03598930 n01844917 n04509417 n02441326 n02786058 n03134739 n03838899 n04192698 n02837789 n02074367 n02701002 n07717070 n03977966 n12992868 n03445777 n04162706 |
| **#14** | n03538406 n03314780 n03916031 n04310018 n04074963 n04462240 n03250847 n01704323 n07753113 n04532106 n09288635 n04033995 n03929855 n03733281 n04562935 n03124043 n03682487 n04487081 n03743016 n03670208 n03980874 n04596742 n03457902 n04536866 n03085013 n03527444 n04099969 n04141975 n04326547 n02825657 |
| **#15** | n04417672 n02966687 n03868242 n02692877 n04435653 n04039381 n02084071 n02776631 n02950826 n04350905 n04552348 n07831146 n04149813 n03787032 n03791053 n04357314 n04476259 n02129604 n03791235 n03992509 n01604330 n03891332 n04613696 n04592741 n02687172 n02782093 n04525038 n02835271 n01674464 n07742313 n02454379 |
| **#16** | n02910353 n02323902 n03327234 n01726692 n03095699 n04443257 n04201297 n02667093 n04584207 n04328186 n02909870 n04311174 n04067472 n04270147 n04344873 n03777754 n03658185 n03706229 n07836838 n03770679 n03208938 n01976146 n02062744 n03697007 n03476684 n02469914 n04458633 n02274259 n10565667 n01872401 n03584254 n04019541 n03461385 |
| **#17** | n03063599 n04576211 n03841143 n03617480 n02992211 n04251144 n04239074 n02131653 n04254120 n02979186 n01514668 n03476991 n04229816 n03776460 n04429376 n01696633 n01905661 n03594945 n04370456 n02159955 n04230808 n03141823 n00001930 n03485407 n04372370 n04285008 n03032252 n04286575 n02894605 n03709823 n02329401 n03160309 n03721384 n03857828 |
| **#18** | n02870880 n03127747 n02880940 n04346328 n04482393 n03800933 n04152593 n03051540 n03042490 n04317175 n03661043 n04548280 n04235860 n02807133 n02790996 n03877472 n07892512 n07871810 n03866082 n07875152 n10148035 n04531098 n03814906 n02927161 n04296562 n03729826 n04023962 n01768244 n00003553 n04127249 n04505470 n03825788 n03794056 n03929660 n03742115 |

*Table 10.* Details of superclasses in ImageNet-1K

| Superclass | Classes of ImageNet | | | | | | | |
|---|---|---|---|---|---|---|---|---|
| **n02084071** | n02085620 | n02085782 | n02085936 | n02086079 | n02086240 | n02086646 | n02086910 | n02087046 |
| | n02087394 | n02088094 | n02088238 | n02088364 | n02088466 | n02088632 | n02089078 | n02089867 |
| | n02089973 | n02090379 | n02090622 | n02090721 | n02091244 | n02091467 | n02091635 | n02091831 |
| | n02092002 | n02092339 | n02093256 | n02093428 | n02093647 | n02093754 | n02093859 | n02093991 |
| | n02094114 | n02094258 | n02094433 | n02095314 | n02095570 | n02095889 | n02096051 | n02096177 |
| | n02096294 | n02096437 | n02096585 | n02097047 | n02097130 | n02097209 | n02097298 | n02097474 |
| | n02097658 | n02098105 | n02098286 | n02098413 | n02099267 | n02099429 | n02099601 | n02099712 |
| | n02099849 | n02100236 | n02100583 | n02100735 | n02100877 | n02101006 | n02101388 | n02101556 |
| | n02102040 | n02102177 | n02102318 | n02102480 | n02102973 | n02104029 | n02104365 | n02105056 |
| | n02105162 | n02105251 | n02105412 | n02105505 | n02105641 | n02105855 | n02106030 | n02106166 |
| | n02106382 | n02106550 | n02106662 | n02107142 | n02107312 | n02107574 | n02107683 | n02107908 |
| | n02108000 | n02108089 | n02108422 | n02108551 | n02108915 | n02109047 | n02109525 | n02109961 |
| | n02110063 | n02110185 | n02110341 | n02110627 | n02110806 | n02110958 | n02111129 | n02111277 |
| | n02111500 | n02111889 | n02112018 | n02112137 | n02112350 | n02112706 | n02113023 | n02113186 |
| | n02113624 | n02113712 | n02113799 | n02113978 | | | | |
| **n01503061** | n01530575 | n01531178 | n01532829 | n01534433 | n01537544 | n01558993 | n01560419 | n01580077 |
| | n01582220 | n01592084 | n01601694 | n01608432 | n01817953 | n01818515 | n01819313 | n01820546 |
| | n01824575 | n01828970 | n01829413 | n01833805 | n01843065 | n01843383 | n02002556 | n02002724 |
| | n02006656 | n02007558 | n02009229 | n02009912 | n02011460 | n02012849 | n02013706 | n02017213 |
| | n02018207 | n02018795 | n02025239 | n02027492 | n02028035 | n02033041 | n02037110 | n02051845 |
| | n02058221 | | | | | | | |
| **n02159955** | n02165105 | n02165456 | n02167151 | n02168699 | n02169497 | n02172182 | n02174001 | n02177972 |
| | n02190166 | n02206856 | n02219486 | n02226429 | n02229544 | n02231487 | n02233338 | n02236044 |
| | n02256656 | n02259212 | n02264363 | | | | | |
| **n02469914** | n02481823 | n02483362 | n02483708 | n02484975 | n02486261 | n02486410 | n02487347 | n02488291 |
| | n02488702 | n02489166 | n02490219 | n02492035 | n02492660 | n02493509 | n02493793 | n02494079 |
| | n02497673 | n02500267 | | | | | | |
| **n01726692** | n01728572 | n01728920 | n01729322 | n01729977 | n01734418 | n01735189 | n01737021 | n01739381 |
| | n01740131 | n01742172 | n01744401 | n01748264 | n01749939 | n01751748 | n01753488 | n01755581 |
| | n01756291 | | | | | | | |
| **n02512053** | n01440764 | n01443537 | n01484850 | n01491361 | n01494475 | n01496331 | n01498041 | n02514041 |
| | n02526121 | n02536864 | n02606052 | n02607072 | n02640242 | n02641379 | n02643566 | n02655020 |
| **n01674464** | n01675722 | n01677366 | n01682714 | n01685808 | n01687978 | n01688243 | n01689811 | n01692333 |
| | n01693334 | n01694178 | n01695060 | | | | | |
| **n02401031** | n02403003 | n02408429 | n02410509 | n02412080 | n02415577 | n02417914 | n02422106 | n02422699 |
| | n02423022 | | | | | | | |
| **n01769347** | n01770081 | n01773157 | n01773549 | n01773797 | n01774384 | n01774750 | n01775062 | n01776313 |
| **n02083346** | n02114367 | n02114548 | n02114712 | n02114855 | n02115641 | n02115913 | n02116738 | n02117135 |
| **n02441326** | n02441942 | n02442845 | n02443114 | n02443484 | n02444819 | n02445715 | n02447366 | |
| **n12992868** | n12985857 | n12998815 | n13037406 | n13040303 | n13044778 | n13052670 | n13054560 | |
| **n02153203** | n01795545 | n01796340 | n01797886 | n01798484 | n01806567 | n01807496 | | |
| **n02120997** | n02125311 | n02127052 | n02128385 | n02128757 | n02128925 | n02130308 | | |
| **n02274259** | n02276258 | n02277742 | n02279972 | n02280649 | n02281406 | n02281787 | | |
| **n04531098** | n02795169 | n02808440 | n03950228 | n04049303 | n04398044 | n04493381 | | |
| **n01629276** | n01629819 | n01630670 | n01631663 | n01632458 | n01632777 | | | |
| **n01662784** | n01664065 | n01665541 | n01667114 | n01667778 | n01669191 | | | |
| **n01905661** | n01924916 | n01950731 | n01955084 | n01990800 | n02321529 | | | |
| **n02121808** | n02123045 | n02123159 | n02123394 | n02123597 | n02124075 | | | |
| **n02329401** | n02342885 | n02346627 | n02361337 | n02363005 | n02364673 | | | |
| **n04341686** | n03781244 | n03788195 | n03837869 | n03877845 | n03956157 | | | |

*Table 11.* Details of superclasses in ImageNet-1K (continued)

| Superclass | Classes of ImageNet | | | | Superclass | Classes of ImageNet | |
|---|---|---|---|---|---|---|---|
| **n01976957** | n01978287 | n01978455 | n01980166 | n01981276 | **n02134971** | n02137549 | n02138441 |
| **n02118333** | n02119022 | n02119789 | n02120079 | n02120505 | **n02268148** | n02268443 | n02268853 |
| **n02131653** | n02132136 | n02133161 | n02134084 | n02134418 | **n03906997** | n02783161 | n03388183 |
| **n04530566** | n02981792 | n03947888 | n04147183 | n04612504 | **n01604330** | n01614925 | n01616318 |
| **n00021265** | n07579787 | n07583066 | n07584110 | n07590611 | **n01696633** | n01697457 | n01698640 |
| **n01639765** | n01641577 | n01644373 | n01644900 | | **n01940736** | n01943899 | n01968897 |
| **n01844917** | n01855032 | n01855672 | n01860187 | | **n01942177** | n01944390 | n01945685 |
| **n01861778** | n01871265 | n02504013 | n02504458 | | **n02062744** | n02066245 | n02071294 |
| **n00002684** | n01914609 | n01917289 | n09256479 | | **n02090827** | n02091032 | n02091134 |
| **n01976146** | n01983481 | n01984695 | n01985128 | | **n02134971** | n02137549 | n02138441 |
| **n02323902** | n02325366 | n02326432 | n02328150 | | **n02268148** | n02268443 | n02268853 |
| **n02395003** | n02395406 | n02396427 | n02397096 | | **n03906997** | n02783161 | n03388183 |
| **n03472232** | n02777292 | n03535780 | n03888605 | | **n03001627** | n02791124 | n03376595 |
| **n03800933** | n02787622 | n02804610 | n03884397 | | **n00001930** | n02799071 | n09835506 |
| **n03791235** | n02814533 | n03100240 | n03930630 | | **n04235291** | n02860847 | n03218198 |
| **n03497657** | n02869837 | n03124170 | n04259630 | | **n04014297** | n02895154 | n03146219 |
| **n03405725** | n03018349 | n03337140 | n04550184 | | **n02883344** | n03014705 | n03127925 |
| **n04576211** | n03272562 | n03393912 | n03895866 | | **n03540267** | n03026506 | n04254777 |
| **n04230808** | n03534580 | n03770439 | n04136333 | | **n03380867** | n03047690 | n03680355 |
| **n02898711** | n04311004 | n04366367 | n04532670 | | **n03682487** | n03075370 | n03874599 |
| **n07707451** | n07714571 | n07716358 | n07718747 | | **n02766320** | n03125729 | n03131574 |
| **n01604330** | n01614925 | n01616318 | | | **n03928116** | n03452741 | n04515003 |
| **n01696633** | n01697457 | n01698640 | | | **n04464852** | n03478589 | n04389033 |
| **n01940736** | n01943899 | n01968897 | | | **n03985232** | n03642806 | n03832673 |
| **n01942177** | n01944390 | n01945685 | | | **n04524313** | n03673027 | n04347754 |
| **n02062744** | n02066245 | n02071294 | | | **n03051540** | n03710637 | n03710721 |
| **n02090827** | n02091032 | n02091134 | | | **n04565375** | n03773504 | n04008634 |
| **n02880940** | n03775546 | n04263257 | | | **n03294048** | n03924679 | n04004767 |
| **n03327234** | n03930313 | n04604644 | | | **n02942699** | n03976467 | n04069434 |
| **n03603722** | n04560804 | n04579145 | | | **n07679356** | n07684084 | n07695742 |
| **n07717070** | n07717410 | n07717556 | | | **n00003553** | n12057211 | n12620546 |
| **n13134947** | n12144580 | n13133613 | | | | | |

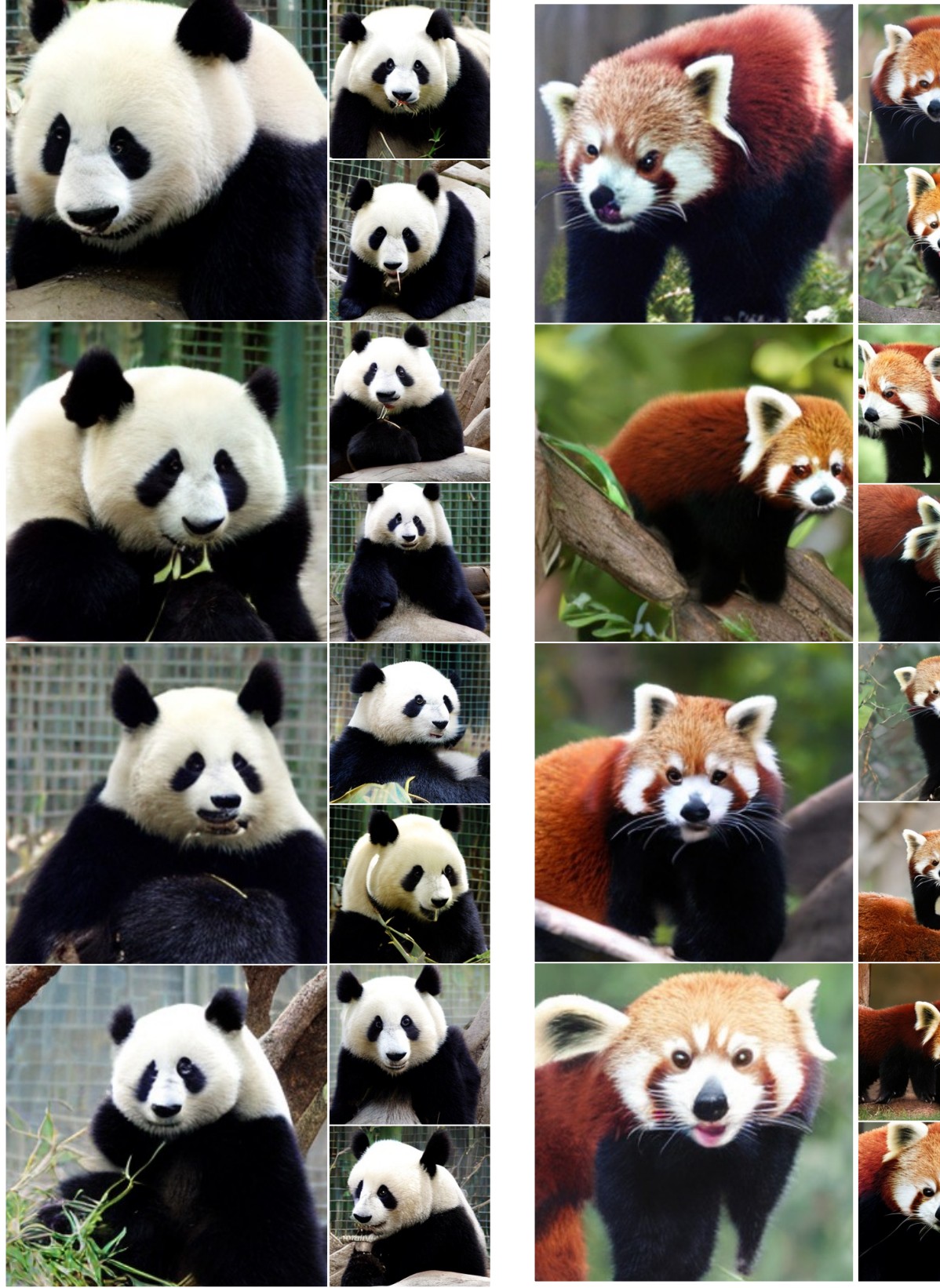

*Figure 8.* Images of n02510455 generated by KIND.

*Figure 9.* Images of n02509815 generated by KIND.

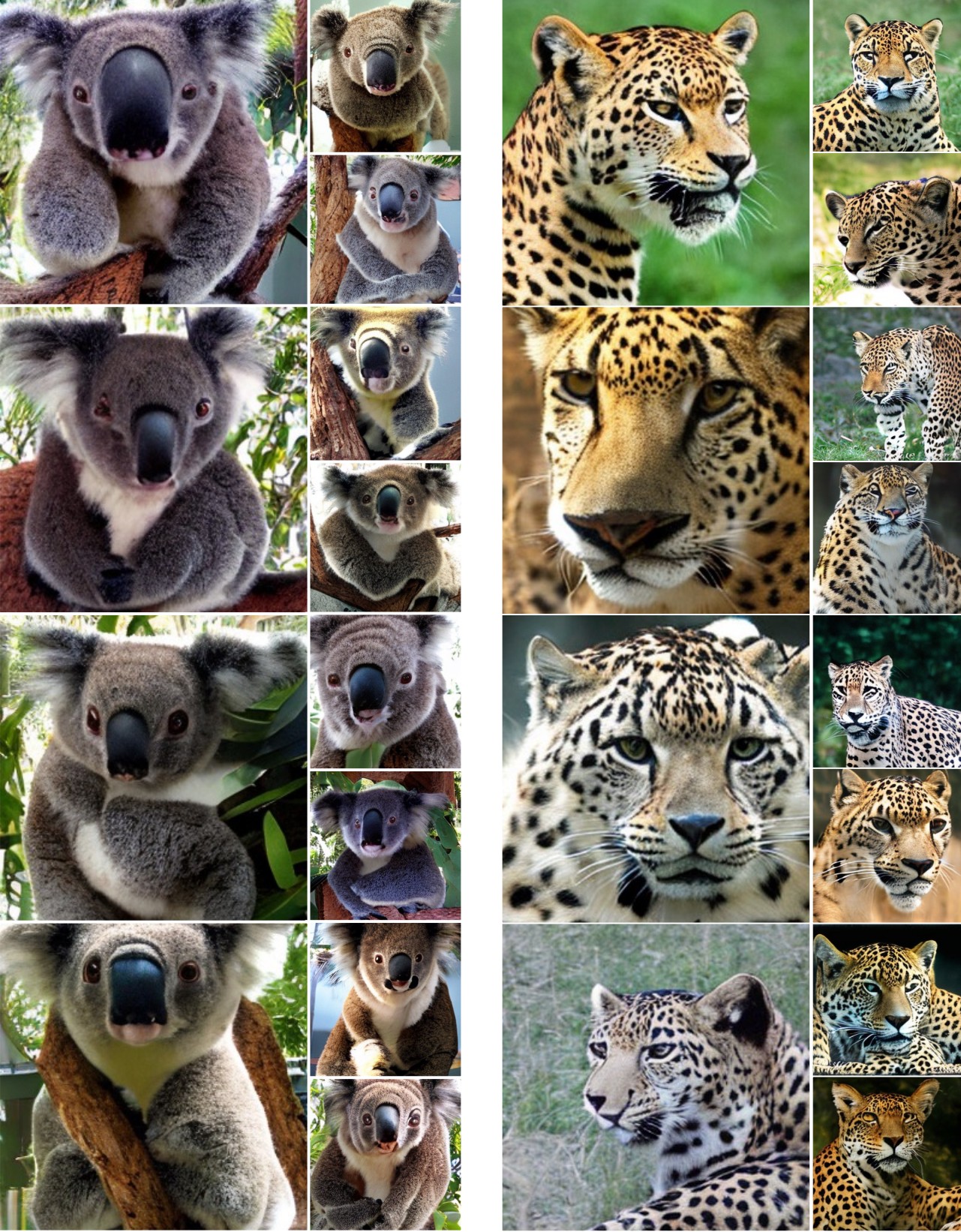

*Figure 10.* Images of n01882714 generated by KIND.

*Figure 11.* Images of n02120997 generated by KIND.

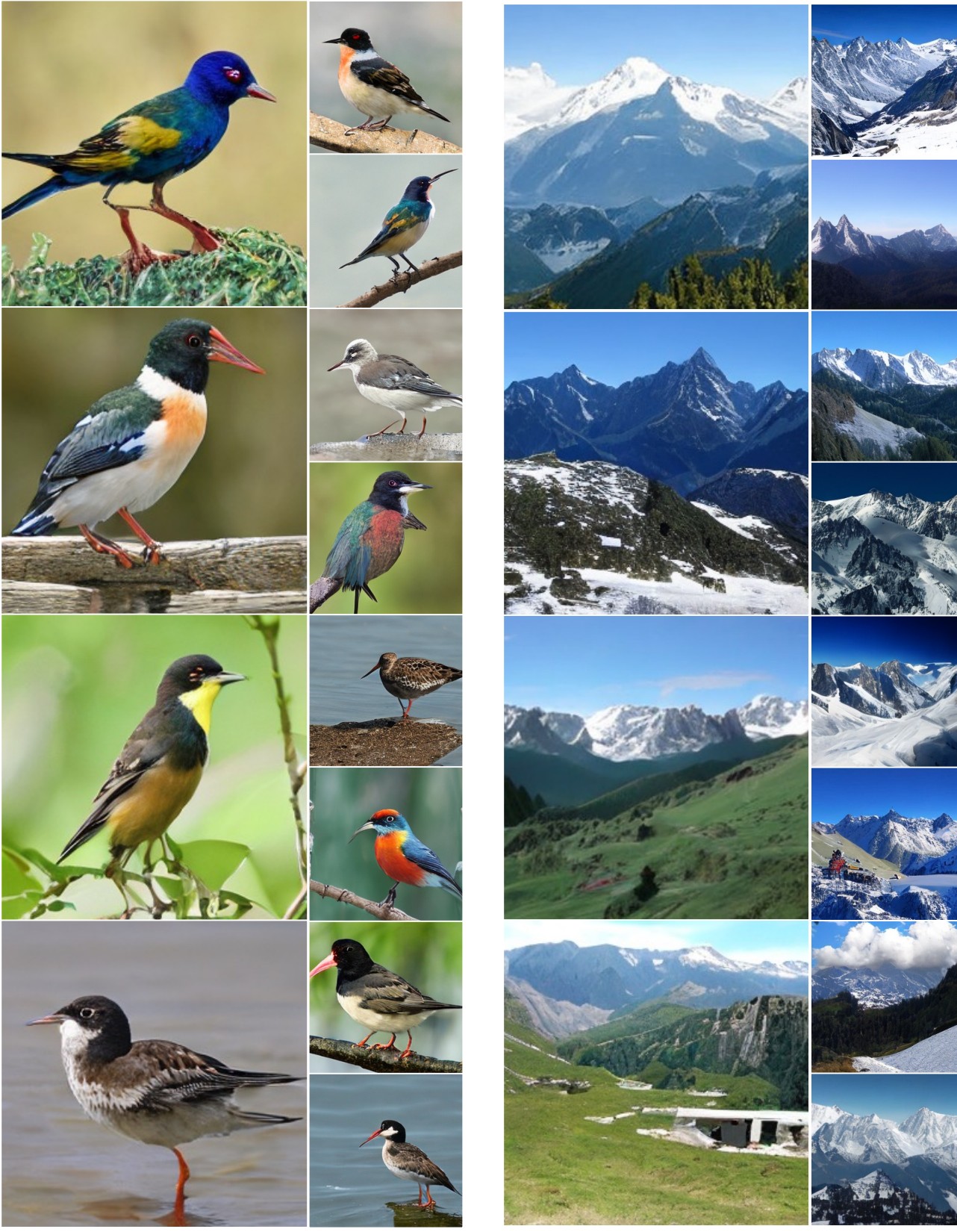

*Figure 12.* Images of n01503061 generated by KIND.

*Figure 13.* Images of n09193705 generated by KIND.

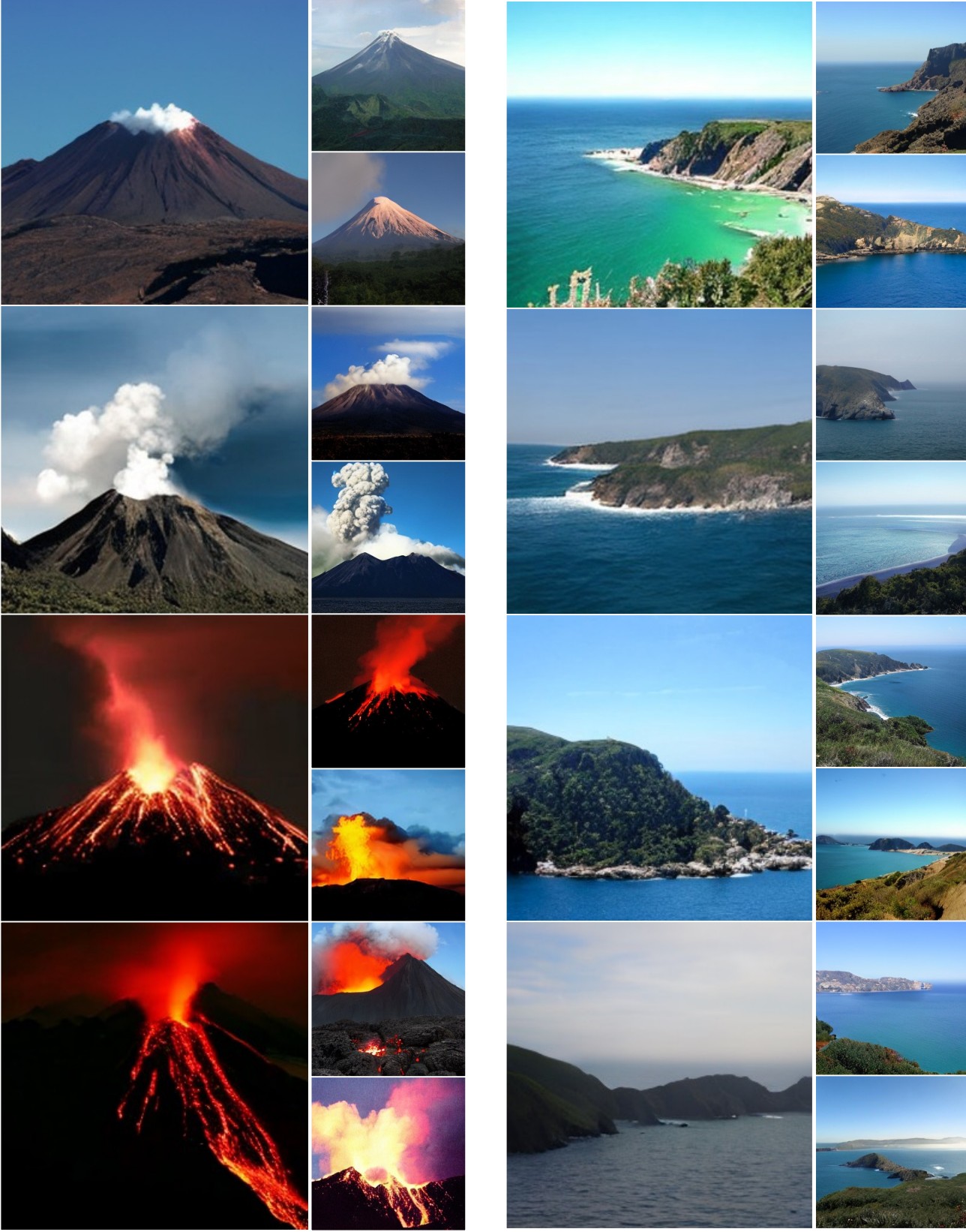

*Figure 14.* Images of n09472597 generated by KIND.

*Figure 15.* Images of n09399592 generated by KIND.

