# OpenReview forum: "KIND: Knowledge Integration and Diversion for Training Decomposable Models"
_ICML.cc/2025/Conference — ICML 2025 poster_

### Official Review · Reviewer_Dcjo · 2025-03-06

**Overall Recommendation:** 3

**Summary:**

This paper tårgets on training a better pre-trained model for downstream tasks. Concretely, they propose KIND (Knowledge Integration and Diversion). It utilizes SVD to yield basic components, and then classify them into two categories, learngenes and tailors. The former captures class-agnostic features, while the latter captures class-specific ones. With SVD, it trains basic components instead of the full-weight matrices. This method is reported to be the first to apply learngenes to image generation tasks. They establish a benchmark for evaluating the transferability of diffusion models. Extensive reported results prove the effectiveness of the proposed method.

## After Rebuttal
After reading the rebuttal and other reviews, I still tend to accept this paper.

**Claims And Evidence:**

One key idea in this method is to adopt SVD to categorize basic components into learngenes and tailors. This claim can be supported by previous works that SVD has been applied to disentangle class-agnostic and class-specific components.

**Essential References Not Discussed:**

N/A from my perspective.

**Experimental Designs Or Analyses:**

I think the experiment section in this paper is good. It covers multiple datasets and settings, and reports enough criteria to validate the performance of methods. I think constructing such benchmark is beneficial to the community.

**Methods And Evaluation Criteria:**

I think the method is reasonable and new to me. Previous methods have applied SVD, but this method also shows enough novelty to me.
I think the benchmark datasets cover some basic datasets in image generation, and plenty of criteria and scenarios have been incorporated in the experiment section.

**Other Comments Or Suggestions:**

N/A

**Other Strengths And Weaknesses:**

The figures in this paper are good, illustrating the idea in a clear manner.

**Questions For Authors:**

I want to see more discussions on the limitations of the proposed method. Some failure cases are also welcome.

**Relation To Broader Scientific Literature:**

I think this is an interesting paper. Besides the interesting method proposed in this paper, the benchmark they construct might be meaningful.

**Theoretical Claims:**

I have checked A3. I have read A2 but I am not familiar with DK Theorem.

---

> ### Author Rebuttal · Authors · 2025-03-30
>
> Dear Reviewer Dcjo,
>
> We sincerely appreciate your insightful comments and your recognition of both the novelty and soundness of our methods, as well as the contribution of our benchmark to the community. Below, we provide a detailed response.
>
> >**Q1: Discussions on the limitations of the proposed method.**
>
> We have briefly discussed the limitations in Section 7 of the original manuscript. Here, we further elaborate on these limitations and outline potential directions for future improvements:
>
>
> - **Limitations in Class-Conditional Generation.**
> To illustrate the transferability of class-agnostic knowledge obtained via knowledge diversion, we focus on class-conditional generation tasks, where variations induced by different class labels naturally introduce downstream tasks with substantial domain shifts.
>
>     In contrast, text-conditional generation, which controls image content via prompts, is also widely adopted.
>     However, large-scale text-to-image diffusion models (e.g., Stable Diffusion 3) are pre-trained on a broad spectrum of internet images, making it challenging to define tasks with significant domain shifts.
>     Moreover, extending KIND to text-conditional generation requires transitioning class gate from discrete, countable class labels to an open-ended, uncountable prompt space, where **a binary class gate is insufficient**.
>
>     Future work will explore Mixture-of-Experts (MoE) techniques for knowledge diversion, enabling dynamic allocation of a limited set of tailors based on prompts.
>     We have conduct preliminary experiments using a PixArt-based model trained for 50,000 steps, see Tabel below. KIND may underperform compared to parameter-efficient fine-tuning (PEFT) methods such as LoRA, as pre-trained text-to-image diffusion models inherently bridge moderate domain differences.
>
>     |Dataset: MRI| CLIP Score↑|LPIPS↓
>     |-|-|-
>     |PixArt-Lora|**33.88**| 0.4252
>     |PixArt-KIND|33.20|**0.4213**
>
>
>     |Dataset: Pokemon|CLIP Score↑|LPIPS↓
>     |-|-|-
>     |PixArt-Lora|32.48|**0.4279**
>     |PixArt-KIND|**32.55**|0.4288
>
> - **Limitations in Structural Expansion.**
>     KIND has been applied exclusively to Transformer-based architectures, focusing on knowledge diversion in Multi-Head Self-Attention ($W_q$, $W_k$, $W_v$, $W_o$) and Pointwise Feedforward layers ($W_{in}$, $W_{out}$).
>     While DiT is becoming the dominant architecture for diffusion models, many classic diffusion models still rely on convolution-based UNets.
>
>     Extending KIND to architectures dominated by convolutional layers presents a key challenge.
>     Although convolutional weights can be represented as three-dimensional tensors and prior work has explored SVD-based decomposition for convolutional layers, the strong inductive biases of convolutional kernels pose unique difficulties.
>     Developing effective knowledge diversion strategies for convolutional networks remains an important direction for future research.
>
> - **Limitations in Other Tasks (e.g., Image Classification).**
>     KIND has been primarily evaluated on image generation tasks, yet its knowledge diversion mechanism—encapsulating class-agnostic and class-specific knowledge into distinct network components—suggests broader applicability.
>
>     A particularly promising direction is cross-domain few-shot learning, where models must generalize across domains with limited data. Traditional methods often struggle under large distribution shifts due to their reliance on prior knowledge from the source domain.
>     KIND offers a key advantage: learngenes serve as a transferable backbone for stable adaptation, while tailors enable task-specific fine-tuning with minimal data, improving generalization.
>
>     However, unlike image generation, image classification lacks access to class labels during inference, requiring each image to traverse all tailors to extract features, leading to increased computational overhead. As the number of classes grows, classification complexity further escalates.
>     Preliminary experiments applying KIND to ViT for cross-domain few-shot learning (see Tabel below) demonstrate significant improvements over baselines, though a performance gap remains compared to state-of-the-art methods. Thus, developing an efficient learngene-tailer framework for classification remains an open research direction.
>
>     ||ChestX|ISIC|EuroSAT|CropDisease|Average
>     |-|-|-|-|-|-
>     |Vanilla ViT (Baseline)|26.3|46.1|88.6|94.6|63.9
>     |P>M>F|**27.3**|50.1|86.0|93.0|64.1
>     |StyleAdv|27.0|**51.2**|**90.1**|**96.0**|**66.1**
>     |KIND|26.5|47.9|88.8|95.0|64.6
>
> We will provide a detailed discussion of these limitations and future research directions in the revised Appendix to inspire further advancements in KIND and broaden its application scope.

---

> > ### Comment · Reviewer_Dcjo · 2025-04-05
> >
> > After reading the rebuttal, most of my concerns are solved. I still tend to accept this paper.

---

### Official Review · Reviewer_Y7yt · 2025-03-10

**Overall Recommendation:** 4

**Summary:**

This manuscript proposes a novel pre-training method named KIND, aiming to address the adaptability issues of traditional pre-trained models in different tasks and deployment scenarios. KIND integrates and distributes knowledge by using SVD during the pre-training process, creating learngenes and tailors respectively, and achieves effective knowledge transfer through a class gate mechanism. Experiments verify that KIND can be flexibly deployed in various scenarios and significantly improve the transfer efficiency in tasks with large domain shifts. The contribution of KIND lies in providing a decomposable structure for pre-trained models, enabling the models to be dynamically adjusted according to task requirements.

## update after rebuttal
The author's response has addressed my concerns, and I am inclined to accept this paper.

**Claims And Evidence:**

The authors analyzed the existing problems and put forward the claim of "rethinking the pre-training process to develop decomposable pre-trained models".

**Essential References Not Discussed:**

The core decomposition method of this paper is quite similar to that of the paper "FacT: Factor-Tuning for Lightweight Adaptation on Vision Transformer" published in AAAI 2023, but the authors did not conduct a comparison and discussion. In addition, the paper "PARAMETER-EFFICIENT ORTHOGONAL FINETUNING VIA BUTTERFLY FACTORIZATION" published in ICLR 2024 should also be compared and discussed.

**Experimental Designs Or Analyses:**

The experimental design is reliable, and the datasets used in the experiments include ImageNet and several downstream datasets.

**Methods And Evaluation Criteria:**

The method is applicable to solving the problem of model decomposition, and the experimental verification indicators are reasonable.

**Other Comments Or Suggestions:**

It is recommended to introduce a README part in the code of the supplementary material, otherwise it will cause inconvenience for reviewers to review.

**Other Strengths And Weaknesses:**

Strengths

1 The overall writing logic is relatively clear.

2 The feature of "no further time-consuming steps required" is very friendly to the environment with limited computing resources.

Weaknesses

1 There is a lack of comparison and discussion of the decomposition methods in FacT of AAAI 2023 and BOFT of ICLR 2024.

2 One of the authors' core arguments is the application in the Limited Resources scenario, but this point is not highlighted in the experimental part. For example, it is recommended to add experiments on mobile devices.

**Questions For Authors:**

The authors compared the number of parameters and the amount of computation in Table 2. It is recommended to supplement the comparison of speed.

**Relation To Broader Scientific Literature:**

The core decomposition method of this paper is quite similar to that of the paper "FacT: Factor-Tuning for Lightweight Adaptation on Vision Transformer" published in AAAI 2023, but no discussion is carried out.

**Theoretical Claims:**

The core theory is based on SVD, which is relatively easy to understand.

---

> ### Author Rebuttal · Authors · 2025-03-30
>
> Dear Reviewer Y7yt,
>
> We sincerely appreciate your insightful feedback and your recognition of the innovation and practicality of our work. Below, we provide our detailed response.
>
> >**Q1: Lack of comparison of similar methods (e.g., FacT and BOFT).**
>
> FacT and BOFT leverage matrix decomposition techniques relevant to this work but are fundamentally Parameter-Efficient Fine-Tuning (PEFT) methods.
> They focus on decomposing pre-trained weight matrices to identify compact parameter subspaces, that can be efficiently fine-tuned for adapting pre-trained models to novel tasks. However, these methods heavily rely on traditional pre-trained models, which are typically fixed in size, structurally inflexible, and risk negative transfer.
>
> In contrast, KIND introduces **a novel pre-training paradigm** that explicitly decomposes knowledge into class-agnostic and class-specific components, encapsulated in learngenes and tailors, respectively. This approach results in decomposable pre-trained models, where the modular design enhances transferability while enabling task-specific adaptation, effectively addressing the limitations of traditional pre-training.
>
> Table 2 in the manuscript highlights the superior transferability of learngenes by comparing KIND with PEFT-based methods, including SVD-based approaches (e.g., SVDiff, PiSSA) related to FacT, and OFT-based methods related to BOFT.
> Per your suggestion, additional comparisons with FacT and BOFT (see Table below) further demonstrate that KIND consistently outperforms these PEFT methods, underscoring its ability to transfer only class-agnostic knowledge while avoiding the deployment challenges and the redundant, biased, or harmful transfer often associated with traditional pre-trained models on which PEFT approaches typically rely.
>
> |DiT-B|CelebA|Hubble|MRI|Pokemon
> |-|-|-|-|-|
> |FacT-TT|0.307|0.242|0.067|0.425
> |BOFT|0.318|0.247|0.058|0.433
> |KIND|**0.201**|**0.124**|**0.042**|**0.343**
>
> |DiT-L|CelebA|Hubble|MRI|Pokemon
> |-|-|-|-|-|
> |FacT-TT|0.240|0.168|0.081|0.299
> |BOFT|0.213|0.155|0.051|0.296
> |KIND|**0.152**|**0.109**|**0.040**|**0.262**
>
> >**Q2: Application in the limited resources scenario is not highlighted in experiments.**
>
> The decomposable model pre-trained by KIND can be flexibly restructured to accommodate computational constraints, enabling efficient deployment on **mobile and edge devices**.
> Although direct evaluation on mobile hardware is beyond our current resources, we approximate its feasibility by analyzing FLOPs, memory footprint, and extrapolating inference latency on modern mobile chip.
>
> We construct a mobile-compatible DiT using KIND and evaluate its efficiency across three state-of-the-art mobile chip: Apple A18, Kirin 9020, and Snapdragon 8 Gen3, which sustain 1907, 1720 and 2774 GFLOPS, respectively.
>
> As shown in Table below, KIND-Mobile achieves a 3.3$\times$ reduction in FLOPs and a 1.8$\times$ reduction in memory usage compared to traditional pre-training, while maintaining strong generative performance (FID=21.14).
> Notably, inference latency remains under 4 seconds across all tested mobile chip, demonstrating KIND’s adaptability in resource-constrained environments.
>
> ||Param.|FLOPs (G)|Memory (MB)|Apple A18 (s)|Kirin 9020 (s)|Snapdragon 8 Gen3 (s)|FID↓|IS↑
> |-|-|-|-|-|-|-|-|-
> |Traditional PT|129.7|43.62|518.8|11.43|12.68|7.86|25.14|47.15
> |KIND-Mobile|**70.2**|**13.22**|**280.8**|**3.47**|**3.84**|**2.38**|**21.14**|**58.18**
>
> >**Q3: Supplement the comparison of speed in Table 2.**
>
> To further emphasize the computational efficiency of KIND on novel tasks, we report the GPU time of different methods in the Table below, following your suggestion.
>
> KIND achieves the best performance with the most efficient training compared with state-of-the-art PEFT methods. This is attributed to its encapsulation of class-agnostic knowledge into learngenes through knowledge diversion, thus enhancing structural flexibility.Notably, under large domain shifts, transferring only learngenes improves the adaptability of the pre-trained model while significantly enhancing transfer efficiency by reducing model parameters through the elimination of redundant class-specific knowledge encapsulated in tailors.
>
> |DiT-B|Para.|FLOPs|GPU Time
> |-|-|-|-
> |SVDiff|**0.1**|43.6|2.5
> |OFT|14.2|119.7|6.6
> |LoRA|12.8|50.1|2.9
> |PiSSA|12.8|50.1|2.9
> |LoHa|12.7|87.1|3.9
> |DoRA|12.8|129.5|6.2
> |Heur-LG|129.6|43.6|4.89
> |Auto-LG|129.6|43.6|4.89
> |Full FT|129.6|43.6|4.89
> |KIND|12.8|**33.7**|**1.6**
>
> |DiT-L|Para.|FLOPs|GPU Time|
> |-|-|-|-
> |SVDiff|**0.2**|155.0|6.6|
> |OFT|50.5|425.6|14.2|
> |LoRA|45.3|178.2|7.1|
> |PiSSA|45.3|178.2|7.1|
> |LoHa|45.3|309.6|12.8|
> |DoRA|45.6|503.0|22.2|
> |Heur-LG|456.8|155.0|10.0|
> |Auto-LG|456.8|155.0|10.0|
> |Full FT|456.8|155.0|10.0|
> |KIND|45.4|**119.6**|**6.3**|
>
> >**Q4: Introduce a README part in the code.**
>
> Thank you for your suggestion. We will include a comprehensive README file in the future open-source release to facilitate the reproduction of KIND.

---

> > ### Comment · Reviewer_Y7yt · 2025-04-03
> >
> > Thank you for the author's reply, which has addressed most of my concerns. The author has added a comparison with similar methods and conducted additional experiments in the mobile scenario.

---

> > > ### Author Response · Authors · 2025-04-03
> > >
> > > Dear Reviewer Y7yt,
> > >
> > > We sincerely appreciate your positive evaluation of our manuscript and your insightful comments. Your detailed feedback has been invaluable in refining our work.
> > >
> > > If there are any remaining concerns, we would be happy to engage in further discussion. Thank you for your time and thoughtful review.
> > >
> > > Best regards.

---

### Official Review · Reviewer_QbMr · 2025-03-12

**Overall Recommendation:** 3

**Summary:**

## Summary
This work applies SVD on the weight matrices $W_q, W_k, W_v, W_o, W_{in}, W_{out}$ of pretrained diffusion transformers (DITs). Then finetunes the SVD decomposed matrices U, $\Sigma, $V$ with target label information. The SVD decomposed matrices are futher splited into two parts to store 1) general information and 2) target label specified information.


## strongthness

- The idea of pretraining DiTs with SVD decomposition of weights is interesting.
- The reconstruction quality of DiTs trained by the proposed method is clearly better than that of other methods. (e.g. Figure 4, 5).


## weakness

- Although the proposed method appears simple (as shown in Algorithm 1), the writing is somewhat hard to follow. I am confused about this paper. For example, line 123  "KIND decomposes pre-trained models" implies applying KIND after pretraining. However, the learning of learngenes and tailors (line 217) requires applying KIND during pretraining. Again, Table 1, 2, 3 compares KIND with other "post-training" methods, while line 251, section 5.1 says "the model pretrained by KIND".

Where does the KIND apply, pretraining or post-training? and What do you want to compare with, pretraining approaches or post-training approaches?

- Assuming KIND is applied on model pretraining (line 251, section 5.1), did other post-training methods in table 1,2,3 use the KIND pretrained DiTs? or normally pretrained DiTs?  Keep in mind that, KIND pretraining benefits from the target label information. Did normally pretrained DiTs, if there is any, use the target label information.

- Without a clarification about the concerns above, the fairness of experimental results is questionable.






## suggestions
- I suggest author to put Algorithm 1 in the main text to make the paper easy to understand. As a reader of an experimental work, I want to know 1) How does the algorithm work (algorithm 1) and 2) What are the results. The Algorithm 1 is so clear and so simple! It should be in the main text.
- Line 150~155 shows the SVD reparameterization is applized on $W_q, W_k, W_v, W_o, W_{in}, W_{out}$. However, Figure 2 implies that the SVD is only applied on $W_{in}, W_{out}$. I suggest to update figure 2 accordingly.

- I am positive on this paper because of the interesting reconstruction quality. However, it is important to have a fair comparison and a clear expression.

### questions

- Algorithm 1 line 1 sayes the initialization of both W and SVD matrices U, $\Sigma$, V. Did you use both matrices, or one? I am confused. It seems that Algorithm 1 line 8 uses SVD matrices U, $\Sigma$, V, rather than W. If this is the case, how do you constrain $U^{\top}U=I$?

**Claims And Evidence:**

check summary

**Essential References Not Discussed:**

no.

**Experimental Designs Or Analyses:**

yes.

**Methods And Evaluation Criteria:**

check summary

**Other Comments Or Suggestions:**

check summary

**Other Strengths And Weaknesses:**

check summary

**Questions For Authors:**

check summary

**Relation To Broader Scientific Literature:**

check summary

**Theoretical Claims:**

no theory.

---

> ### Author Rebuttal · Authors · 2025-03-29
>
> Dear Reviewer QbMr,
>
> We sincerely appreciate your valuable comments and recognition of our work’s innovation and performance. Below is our detailed response.
>
> >**Q1: Where does the KIND apply, pretraining or post-training? What do you want to compare with, pretraining approaches or post-training approaches?**
>
> KIND is a novel pre-training method for constructing decomposable models. Unlike traditional pre-training approaches, KIND explicitly partitions knowledge into class-agnostic and class-specific components, encapsulating them in learngenes and tailors, respectively.
> This decomposition transforms fixed-size models into modular structures, where learngenes enhance transferability, and tailors adapt to specific tasks.
>
> KIND operates during pre-training, producing learngenes and tailors for flexible downstream deployment.
> Accordingly, we first compare it with pre-training methods such as traditional PT and Laptop-Diff, demonstrating that training a decomposable model incurs no extra computational cost while improving structural flexibility.
> To highlight the strong transferability of class-agnostic knowledge, we also compare it with parameter-efficient fine-tuning methods (i.e., post-training approaches) such as LoRA and PiSSA, particularly in tasks with large domain shifts.
>
> In summary, KIND is a pre-training framework that enhances post-training adaptability by enabling adaptive model scaling based on task demands and computational resources.
>
> >**Q2: Did other post-training methods in Table 1,2,3 use the KIND pretrained DiTs or normally pretrained DiTs? Did normally pretrained DiTs use the target label information?**
>
> As noted in Q1, KIND is a pre-training framework for decomposable models that enhances post-training flexibility.
> Table 1 compares different pre-training strategies, including traditional PT and knowledge distillation (e.g., Laptop-Diff), in constructing models of various sizes.
> In contrast, Tables 2 and 3 evaluate downstream performance, comparing KIND with parameter-efficient fine-tuning methods applied to **normally pretrained DiTs**.
>
> This setup ensures a fair comparison, as normally pretrained DiTs also incorporate target label information.
> Normally pretrained DiTs generate images based on class-conditioned information and inherently depend on class labels during pre-training.
> To further validate fairness, we compare the training performance of KIND-pretrained models with normally pretrained models (see Table below).
> The results confirm that normally pretrained models perform comparably to KIND, demonstrating that improvements in Tables 2 and 3 solely stem from the class-agnostic knowledge encapsulated in learngenes.
>
> ||Model|Steps|FID↓|sFID↓|IS↑|Prec.↑|Rec.↑
> |-|-|-|-|-|-|-|-
> |Traditional PT|DiT-L|300K|9.68|**6.15**|72.22|0.69|**0.47**
> |KIND|DiT-L|300K|**9.33**|6.80|**79.39**|0.69|0.46
>
> >**Q3: Fairness of experimental results.**
>
> We have addressed your concerns in detail in Q1 and Q2, which we believe sufficiently clarify this issue.
>
> >**Q4: Algorithm 1 line 1 says the initialization of both $W$ and SVD matrices $U$, $\Sigma$, $V$. Did you use both matrices, or one? How do you constrain $U^\top U=I$?**
>
> We apologize for any confusion caused by the imprecise wording in Algorithm 1, line 1.
> As you noted, during knowledge diversion, gradient updates are applied only to $U$, $\Sigma$ and $V$, while $W$ is indirectly updated via Eq. (5) as $W=U\Sigma V^\top$.
>
> Regarding the constraint $U^\top U=I$, we initially explored enforcing orthogonality using **Cayley parameterization** (details can be found in official PyTorch documentation), a transformation that maps a skew-symmetric matrix to an orthogonal matrix.
> Specifically, we can construct $U$ as $U=(I+Q)(I-Q)^{-1}$, where $Q$ is a skew-symmetric matrix satisfying $Q=-Q^\top$.
> While this guarantees orthogonality, it incurs substantial computational overhead ($\sim7\times$, see table below) due to the matrix inversion, without notable empirical benefits.
>
> ||Model|Steps|GPU Time|FID↓|sFID↓|IS↑|Prec.↑|Rec.↑
> |-|-|-|-|-|-|-|-|-
> |w/ Cayley|DiT-B/4|200K|104.8 hour|56.99|**47.04**|24.7|0.38|**0.46**
> |w/o Cayley|DiT-B/4|200K|14.7 hour|**52.78**|49.66|**25.7**|**0.40**|0.45
>
> Given these trade-offs, we do not enforce explicit orthogonality constraints. Instead, we employ class gates to associate distinct feature representations with corresponding singular vectors, thereby naturally mitigating correlations among the singular vectors. This enables $U$ and $V$ to approximate orthogonality through the learning process.
>
> As shown in [Figure](https://anonymous.4open.science/api/repo/a-8112/file/12.pdf), the process of knowledge diversion through class gate allows the learned matrices to preserve orthogonality without requiring explicit constraints.
>
> >**Q5: Other suggestions.**
>
> Thank you for your valuable suggestion. We will move Algorithm 1 to the main text and complete the missing details in Figure 2 in revision.

---

### Official Review · Reviewer_abHL · 2025-03-14

**Overall Recommendation:** 3

**Summary:**

This paper proposes a method to decompose a model’s learnable matrices into class-agnostic knowledge (learngenes) and class-specific knowledge (tailors) using Singular Value Decomposition (SVD). The learning process for tailors is regulated by a class gate, ensuring that only one class is activated per image. After training, the decomposed components can be flexibly recombined based on specific downstream tasks and resource constraints by selecting the required tailors.

The authors conduct experiments using a generative DiT model to demonstrate a better trade-off between the number of training parameters and generation quality. Additionally, the proposed method improves knowledge transferability to novel classes and datasets with larger domain shifts.

**Claims And Evidence:**

•  The claimed benefit of reduced training complexity (Lines 257–258) lacks sufficient experimental support. If training classes are isolated using the class gate, does this introduce sparse gradients, potentially making the training process more challenging?

•  The claim that learngenes capture task-agnostic information is not well supported by the visualization in Figure 7. The generated images from learngenes exhibit recognizable object-like patterns, and different seeds produce distinct patterns. This observation contradicts the assertion that learngenes do not favor any specific class. Further clarification or empirical validation is needed to reconcile this discrepancy.

•  The extent of class-agnostic information varies across different classes. For example, fur may be considered class-agnostic when training involves only cats and dogs, but this assumption may not hold in a more diverse dataset. This suggests that the knowledge encoded in learngenes is highly dependent on the specific combination of training classes. This dependency should be explicitly discussed.

•  While learngenes are expected to be transferable across different domains, the exact nature of the transferred knowledge remains unclear. The paper does not provide a thorough validation of what aspects of knowledge are being effectively transferred. Additional experiments or analysis would help substantiate this claim.

•  A cross-class validation is missing to support the claim that tailors are class-dependent. For instance, what happens if the tailors from class A are used to generate images for class B? Including such an experiment would help confirm whether tailors truly capture class-specific information and whether their utility is restricted to the classes they were trained on.

**Essential References Not Discussed:**

The authors are encouraged to discuss some related papers in the area of NAS, which train one model but can be regrouped at no cost at inference [A]

[A] Once-for-All: Train One Network and Specialize it for Efficient Deployment. ICLR'20.

**Experimental Designs Or Analyses:**

•  What happens when training with a significantly larger number of classes? Since the parameter size is directly linked to the size of tailors, as shown in Table 1, will the model expand and scale linearly? If so, does this suggest that the proposed method’s efficiency gains diminish as the number of classes increases? Clarifying the scalability of the approach would strengthen the discussion.

•  The details for Table 5 are missing. Additional explanations regarding the setup, evaluation metrics, and key findings should be provided to ensure clarity and completeness.

**Methods And Evaluation Criteria:**

See the Claims And Evidence session.

**Other Comments Or Suggestions:**

N/A

**Other Strengths And Weaknesses:**

see previous sessions.

**Questions For Authors:**

•  For tasks involving novel classes, how is the tailor initialized? Is the entire tailor randomly initialized, or only a portion of it? It is assumed that each task begins with an initialized tailor, but since different tasks involve varying numbers of classes, does this imply that the model size is variable? The parameter values reported in Table 2 appear to be fixed, clarification on this aspect is needed.

•  Is there a direct relationship between the number of training parameters and FLOPS? If so, providing explicit details or empirical validation would strengthen the discussion.

•  What is the exact improvement of KIND over FT in Table 3? Reporting the numerical difference would make the performance comparison more precise and informative.

•  Can this method generate combinations of multiple classes? If so, how is the combination controlled or influenced by the tailors? If not, what are the key limitations preventing such functionality?

•  How are closely related tailors selected for fine-tuning, as described in Line 217? What is the selection criterion, and how many tailors are chosen for fine-tuning? A more detailed explanation of this process would improve clarity and reproducibility.

**Relation To Broader Scientific Literature:**

The key contributions are related to knowledge decomposition, parameter decomposition, and recombination. They might also relate to research areas such as domain shift and model personalization.

**Theoretical Claims:**

The core assumption in Line 636 states that the learned weight is close to an underlying matrix and that the matrix decomposition holds when perturbations are sufficiently small. How can this assumption be systematically evaluated across different scenarios? In tasks with larger domain gaps, the distinction between learngenes and tailors becomes more pronounced. How can this difference be quantified to ensure that the assumption remains valid? Providing empirical or theoretical justification for this assumption across varying domain shifts would strengthen the paper’s argument.

---

> ### Author Rebuttal · Authors · 2025-03-28
>
> Dear Reviewer abHL,
>
> We sincerely appreciate your recognition of our practicality and efficiency.
> Due to length constraints, **experimental tables and figures, are provided via anonymous links (permitted by ICML25)**.
>
> >**Q1:Class Gate and Sparse Gradients**
>
> Parameter updates remain sufficient without excessive sparsity (see [Table](https://anonymous.4open.science/api/repo/a-8112/file/1.pdf)). The learngene, shared across all classes, receives updates from all samples, while tailors benefit from batch training, ensuring broad class coverage (batch size=256, total classes=150).
>
> This setup also reduces gradient conflicts across classes, allowing KIND to achieve competitive performance with similar overhead while maintaining a flexible, decomposable structure.
>
> >**Q2:Task-agnostic Knowledge in Learngenes**
>
> Figure 7 shows that images generated solely from learngenes lack class-specific semantics (visually similar regardless of input labels under the same seed).
>
> To quantify this, [Table](https://anonymous.4open.science/api/repo/a-8112/file/2.pdf) (i.e., Table 5) analyzes the classification distributions of InceptionNet across raw ImageNet images, pre-trained model outputs, learngene-generated images, and pure noise. Entropy, variance, and kurtosis are used to measure distribution uniformity, discreteness, and sharpness.
>
> Results show that learngene-generated images exhibit low correlation with all ImageNet classes and align statistically closer to noise, confirming their class-agnostic nature and lack of semantics.
>
> >**Q3:Specific Combinations of Training Class**
>
> This concern arises only with extremely limited classes, e.g., shared features like fur may dominate between cats and dogs, but introducing a distinct class, such as turtles, reduces this effect.
>
> [Tabel](https://anonymous.4open.science/api/repo/a-8112/file/3.pdf) shows that beyond 100 classes, additional classes offer minimal benefit. As long as the class set is sufficiently diverse, its specific composition has little impact.
>
> >**Q4:Nature of Transferred Knowledge**
>
> BLIP-based VQA verification in [Tabel](https://anonymous.4open.science/api/repo/a-8112/file/4.pdf) confirms that learngene-generated images are natural images (only 5.1\% are classified as 'noisy images').
> This suggests that learngenes encode a general noise-to-image mapping, while tailors inject class semantics.
>
> >**Q5:Cross-class Validation**
>
> As shown in [Figure](https://anonymous.4open.science/api/repo/a-8112/file/5.pdf), the 'panda' tailor fails to generate images for other classes, confirming its class-specific nature.
>
> >**Q6: Theoretical Assumptions**
>
> The core assumption holds under large domain shifts.
> Frobenius norm in [Tabel](https://anonymous.4open.science/api/repo/a-8112/file/6.pdf) shows minimal weight perturbations ($||E^{[t]}||\ll||W^*||$) when transferring learngenes across domains.
>
> >**Q7:Larger Number of Classes**
>
> A larger number of training classes is unnecessary, as 100 classes is sufficient for capturing class-agnostic knowledge (see Q3).
> Once the class-agnostic knowledge has encapsulated in learngenes, additional classes can be integrated by training only corresponding tailors.
>
> >**Q8:Missing Details for Table 5**
>
> See Q2.
>
> >**Q9:Related Works in NAS**
>
> Both KIND and NAS support variable-sized models, but only KIND extracts class-agnostic knowledge for transferability, while NAS focuses solely on network structure.
>
> [Tabel](https://anonymous.4open.science/api/repo/a-8112/file/7.pdf) shows that NAS struggles with domain shifts.
>
> >**Q10:Tailor Initialization and Model Size**
>
> In new tasks, the tailor is typically randomly initialized, which is simple and effective (see Q14).
>
> The number of tailors and parameters varies with class count, with Table 2 reporting the **average parameters** across tasks.
>
> Model size is also influenced by task complexity, and adjusting each tailor's rank balances performance and efficiency (see [Tabel](https://anonymous.4open.science/api/repo/a-8112/file/8.pdf)).
>
> >**Q11:Relationship between Parameters and FLOPs**
>
> Training parameters and FLOPs are related but not strictly proportional, as FLOPs depend on both parameter count and computational patterns like matrix multiplications.
>
> >**Q12:Performance Gains over FT**
>
> We further compare full-parameter fine-tuning in [Tabel](https://anonymous.4open.science/api/repo/a-8112/file/9.pdf), confirming that KIND improves performance with lower computational cost.
>
> >**Q13:Multi-class Generation**
>
> Multi-class generation can be achieved by setting the class gate to 1 for desired classes (see [Figure](https://anonymous.4open.science/api/repo/a-8112/file/10.pdf)).
>
> >**Q14:Selection Criterion for Tailors**
>
> As noted in Q10, randomly initializing tailors is simple and effective.
> Future work may explore fine-tuning from similar-class tailors (e.g., Koala from Monkey in [Tabel](https://anonymous.4open.science/api/repo/a-8112/file/11.pdf)) or adaptively integrating multiple tailors via MoE.

---

> > ### Comment · Reviewer_abHL · 2025-04-03
> >
> > Thank the authors for the rebuttal. Most of my concerns have been resolved. I have updated my rating.

---

> > > ### Author Response · Authors · 2025-04-03
> > >
> > > Dear Reviewer abHL,
> > >
> > > Thank you for your thoughtful evaluation, as well as for your generous score adjustment.
> > >
> > > Your insightful feedback has been instrumental in refining our study, and we sincerely appreciate the time and effort you have dedicated to the review process.
> > >
> > > Once again, we extend our deepest gratitude for your valuable comments!
> > >
> > > Best regards!

---

### Decision · Program_Chairs · 2025-05-01

**Decision:**

Accept (poster)

**Comment:**

This paper introduces KIND, a pretraining method that decomposes model weights via SVD into class agnostic learngenes and class specific tailors, which enables modular adaptation and efficient transfer. KIND is particularly effective under domain shifts, where transferring only the learngenes yields strong performance with minimal overhead. Reviewers found the method technically sound and empirically strong, and the rebuttal addressed remaining key concerns around method's clarity, fairness of the comparison, and related work. Yet, there remains some limitations, including its narrow focus on transformer-based generative models, unknown scalability with many classes, and limited evaluation beyond vision tasks. Still, I found the the overall contribution as solid and impactful and recommend acceptance.